# Bayesian Optimisation of Functions on Graphs

**Xingchen Wan**[*]**, Pierre Osselin**[*]**, Henry Kenlay**
**Binxin Ru, Michael A. Osborne, Xiaowen Dong**
Department of Engineering Science, University of Oxford
{xwan,osselinp,kenlay,robin,mosb,xdong}@robots.ox.ac.uk

## Abstract

The increasing availability of graph-structured data motivates the task of optimising over functions defined on the node set of graphs. Traditional graph search algorithms can be applied in this case, but they may be sample-inefficient and do not make use of information about the function values; on the other hand, Bayesian optimisation is a class of promising black-box solvers with superior sample efficiency, but it has scarcely been applied to such novel setups. To fill this gap, we propose a novel Bayesian optimisation framework that optimises over functions defined on generic, large-scale and potentially unknown graphs. Through the learning of suitable kernels on graphs, our framework has the advantage of adapting to the behaviour of the target function. The local modelling approach further guarantees the efficiency of our method. Extensive experiments on both synthetic and real-world graphs demonstrate the effectiveness of the proposed optimisation framework.

## 1 Introduction

Data collected in a network environment, such as transportation, financial, social, and biological networks, have become pervasive in modern data analysis and processing tasks. Mathematically, such data can be modelled as functions defined on the node set of graphs that represent the networks. This then poses a new type of optimisation problem over functions on graphs, i.e. searching for the node that possesses the most extreme value of the function. Real-world examples of such optimisation tasks are abundant. For instance, if the function measures the amount of delay at different locations in an infrastructure network, one may think about identifying network bottlenecks; if it measures the amount of influencing power users have in a social network platform, one may be interested in finding the most influential users; if it measures the time when individuals were infected in an epidemiological contact network, an important task would be to identify "patient zero" of the disease.

Optimisation of functions on graphs is challenging. Graphs are an example of discrete domains, and conventional algorithms, which are mainly designed for continuous spaces, do not apply straightforwardly. Real-world graphs are often extremely large and sometimes may not even be fully observable. Finally, the target function, such as in the examples given above, is often a black-box function that is expensive to evaluate at the node level and may exhibit complex behaviour on the graph.

Traditional methods to traverse the graph, such as breadth-first search (BFS) or depth-first search (DFS) [14], are heuristics that may be adopted in this setting for small-scale graphs, but inefficient to deal with large-scale real-world graphs and complex functions. Furthermore, these search methods only rely on the graph topology and ignore the function on the graph, which can be exploited to make the search more efficient. On the other hand, Bayesian optimisation (BO) [16] is a sample-efficient sequential optimisation technique with proven successes in various domains and is suitable for solving black-box, expensive-to-evaluate optimisation problems. However, while BO has been combined with graph-related settings, e.g. optimising for *graph structures* (i.e. the *individual configurations*

---

[*]Equal contribution.

37th Conference on Neural Information Processing Systems (NeurIPS 2023).

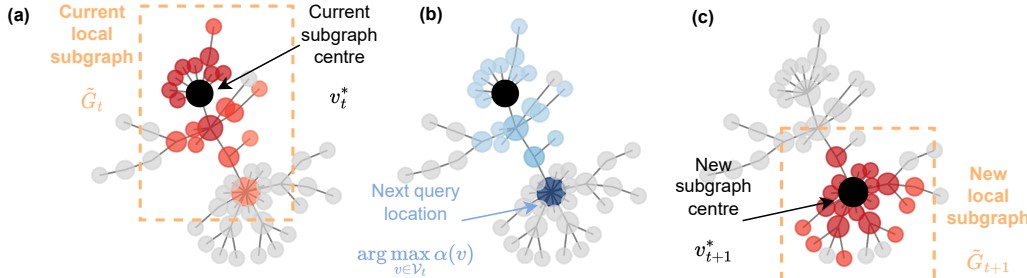

Figure 1: Illustration of one iteration of *BayesOptG* on an example graph. **(a)** At iteration $t$, we construct a local subgraph $\tilde{G}_t$ centred around $v_t^*$ whose nodes are marked in orange-red, with darker shade denoting a shorter distance to $v_t^*$, the best node seen so far (marked in black), and nodes outside $\tilde{G}_t$ are marked in grey. The readers are referred to §3.2 for the details; **(b)** we place a GP surrogate with the covariance function defined in §3.1 on $\tilde{G}_t$ and pick the maximiser of the acquisition function (the acquisition function values are marked in shades of blue, with a darker shade denoting a higher acquisition value) as the node to query for iteration $t + 1$ ($v_{t+1}$) (§3.2) and **(c)** if querying $v_{t+1}$ leads to a better objective function value ($f(v_{t+1}) < f(v_t^*)$, assuming minimisation), the neighbourhood around it is selected as the new subgraph $\tilde{G}_{t+1}$. The process continues until convergence or a pre-set number of evaluations is reached.

that we optimise for are graphs) in the context of neural architecture search [20, 34], graph adversarial examples [42] or molecule design [23], it has not been applied to the problem of optimising over functions on graphs (i.e. the *search space* is a graph and the configurations we optimise for are *nodes* in the graph). The closest attempt was COMBO [27], which is a framework designed for a specific purpose, i.e. combinatorial optimisation, where the search space is modelled as a synthetic graph restricted to one that can be expressed as a Cartesian product of subgraphs. It also assumes that the graph structure is available and that the function values are smooth in the graph space to facilitate using a diffusion kernel. All these assumptions may not hold in the case of optimisation over generic functions on real-world graphs.

We address these limitations in our work, and our main contributions are as follows: we consider the problem setting of optimising functions that are supported by the node set of a potentially generic, large-scale, and potentially unknown graph – *a setup that is by itself novel* to the best of our knowledge in the BO literature. We then propose a novel BO framework that effectively optimises in such a problem domain with 1) appropriate kernels to handle the aforementioned graph search space derived by spectral learning on the local subgraph structure and is therefore flexible in terms of adapting to the behaviour of the target function, and 2) efficient local modelling to handle the challenges that the graphs in question can be large and/or not completely known a-priori. Finally, we deploy our method in various novel optimisation tasks on both synthetic and real-world graphs and demonstrate that it achieves very competitive results against baselines.

## 2 Preliminaries

BO is a zeroth-order (i.e. gradient-free) and sample-efficient sequential optimisation algorithm that aims to find the global optimum $x^*$ of a black-box function defined over search space $\mathcal{X}$: $x^* = \arg\min_{x \in \mathcal{X}} f(x)$ (we consider a minimisation problem without loss of generality). BO uses a statistical surrogate model to approximate the objective function and an acquisition function $\alpha(x)$ to balance exploitation and exploration under the principle of optimism in the face of uncertainty. At the $t$-th iteration of BO, the objective function is queried with a configuration $x_t$ and returns an output $y_t$, a potentially noisy estimator of the objective function $y_t = f(x_t) + \epsilon, \epsilon \sim \mathcal{N}(0, \sigma_n^2)$ where $\sigma_n^2$ is the noise variance. The statistical surrogate is trained on the observed data up to $t$-th observation $\mathcal{D}_t = \{(x_i, y_i)\}_{i=1}^t$ to approximate the objective function. In this work, we use a Gaussian process (GP) surrogate, which is query-efficient and gives analytic posterior mean and variance estimates on the unknown configurations. Formally, a GP is denoted as $f(x) \sim \text{GP}(m(x), k(x, x'))$, where $m(x)$ and $k(x, x')$ are the mean function and the covariance function (or the *kernel*), respectively. While the mean function is often set to zero or a simple function, the covariance function encodes our belief on the property of the function we would like to model, the choice of which is a crucial design decision when using GP. The covariance function typically has some kernel hyperparameters $\theta$ and are typically optimised by maximising the *log-marginal*

*likelihood* (the readers are referred to detailed derivations in Rasmussen [31]). With $m(\cdot)$ and $k(\cdot, \cdot)$ defined, at iteration $t$, with $\mathbf{X}_t = [x_1, ..., x_t]^\top$ and the corresponding output vector $\mathbf{y}_{1:t} = [y_1, ..., y_t]^\top$, a GP gives analytic posterior mean $\mu(x_{t+1}|\mathcal{D}_t) = \mathbf{k}(x_{t+1}, \mathbf{X}_{1:t})\mathbf{K}_{1:t}^{-1}\mathbf{y}_{1:t}$ and variance $k(x_{t+1}, x'_{t+1}|\mathcal{D}_t) = k(x_{t+1}, x'_{t+1}) - \mathbf{k}(x_{t+1}, \mathbf{X}_{1:t})\mathbf{K}_{1:t}^{-1}\mathbf{k}(\mathbf{X}_{1:t}, x'_{t+1}))$ estimates on an unseen configuration $x_{t+1}$, where $[\mathbf{K}_{1:t}]_{i,j} = k(x_i, x_j)$ is the $(i, j)$-th element of the Gram matrix induced on the $(i, j)$-th training samples by $k(\cdot, \cdot)$, the covariance function. With the posterior mean and variance predictions, the acquisition function is optimised at each iteration to recommend the configuration (or a batch of configurations for the case of batch BO) to be evaluated for the $t + 1$-th iteration. For additional details of BO, the readers are referred to Frazier [15].

## 3   Bayesian Optimisation on Graphs

**Problem setting.**   Formally, we consider a novel setup with a graph $G$ defined by $(\mathcal{V}, \mathcal{E})$, where $\mathcal{V} = \{v_i\}_{i=1}^n$ are the nodes and $\mathcal{E} = \{e_k\}_{k=1}^m$ are the edges where each edge $e_k = \{v_{i'}, v_{j'}\}$ connects nodes $v_{i'}$ and $v_{j'}$. The topology $G$ may be succinctly represented by an adjacency matrix $\mathbf{A} \in \{0, 1\}^{n \times n}$; in our case, $m$ and $n$ are potentially large, and the overall topology is not necessarily fully revealed to the search algorithm at running time. It is worth noting that, for simplicity, we focus on the setup of *undirected, unweighted* graph where elements of $\mathbf{A}$ are binary and symmetrical (i.e. $A_{ij} = A_{ji}$)[2]. Specifically, we aim to optimise the black-box, typically expensive objective function that is defined *over the nodes*, i.e. it assigns a scalar value to each node in the graph. In other words, the search space (i.e. $\mathcal{X}$ in §2) in our setup is the set of nodes $\mathcal{V}$ and the goal of the optimisation problem is to find the configuration(s) (i.e. $x$ in §2) that minimise the objective function $v^* = \arg\min_{v \in \mathcal{V}} f(v)$.

**Promises and challenges of BO on graphs.**   We argue that BO is particularly appealing under the described setup as (1) it is known to be query-efficient, making it suitable for optimising expensive functions, and (2) it is fully black-box and gradient-free; indeed, we often can only observe inputs and outputs of many real-world functions, and gradients may not even exist in a practical setup. However, there exist various challenges in our setup that make the adaptation of BO highly non-trivial, and despite the prevalence of problems that may be modelled as such and the successes of BO, it has not been extended to the optimisation of functions on graphs. Some examples of such challenges are:

(i) **Exotic search space.** BO is conventionally applied in continuous Euclidean spaces, whereas we focus on discrete graph search spaces. The differences in search space imply that key notions to BO, such as the similarity between two configurations and expected smoothness of objective functions (the latter is often used as a key criterion in selecting the covariance function to use), could differ significantly. For example, while comparing the similarity between two points in a Euclidean space requires only the computation of simple distance metrics (like $\ell_2$ distance), careful thinking is required to achieve the same in comparing two nodes in a graph that additionally accounts for the topological properties of the graph.

(ii) **Scalability.** Real-world graphs such as citation and social networks can often feature a very large number of nodes while not presenting convenient properties such as the graph Cartesian product assumption in Oh et al. [27] to accelerate computations. Therefore, it is a technical challenge to adapt BO in this setting while still retaining computational tractability.

(iii) **Imperfect knowledge on the graph structure.** Related to the previous point, it may also be prohibitively expensive or even impossible to obtain perfect, complete knowledge on real-world graphs beforehand or at any point during optimisation (e.g. obtaining the full contact tracing graph for epidemiology modelling); as such, any prospective method should be able to handle the situation where the graph structure is only revealed incrementally, on-the-fly.

**Overview of *BayesOptG*.**   To effectively address these challenges while retaining the desirable properties of BO, we propose to extend BO to this novel setup and are, to the best of our knowledge, the first to do so. To achieve that, we propose Bayesian Optimisation on Graphs, or ***BayesOptG*** in short, and an illustration of the overall procedure is shown in Fig. 1, and an algorithmic description is available in Algorithm 1. For the rest of this section, we discuss in detail the key components of *BayesOptG* and how the method specifically addresses the challenges identified above.

---

[2]We note that it is possible to extend the proposed method to more complex cases by using the corresponding definitions of Laplacian matrix. We defer thorough analysis to future work.

### 3.1 Kernels for BO on Graphs

**Kernel design.** Covariance functions are crucial to GP-based BO. To use BO in our setup, a covariance function that gives a principled similarity measure between two nodes $\{v_i, v_j\} \subseteq \mathcal{V}$ is required to interpolate signals on the graph effectively. In this paper, we study several kernels, including both those proposed in the literature (e.g. the *diffusion kernel on graphs* and the *graph Matérn kernel* [4]) and two novel kernels designed by us. Following Smola & Kondor [37], all the kernels investigated can be considered in a general formulation. Formally, for a generic graph $\tilde{G} = (\tilde{\mathcal{V}}, \tilde{\mathcal{E}})$ with $\tilde{n}$ nodes and $\tilde{m}$ edges, we define $\tilde{\mathbf{L}} := \frac{1}{2}\left(\mathbf{I} - \tilde{\mathbf{D}}^{-\frac{1}{2}}\tilde{\mathbf{A}}\tilde{\mathbf{D}}^{-\frac{1}{2}}\right)$, where $\mathbf{I}$ is the identity matrix of order $\tilde{n}$, $\tilde{\mathbf{A}}$ and $\tilde{\mathbf{D}}$ are the *adjacency matrix* and the *degree matrix* of $\tilde{G}$, respectively (the term after $\frac{1}{2}$ is known as the *normalised Laplacian matrix* with eigenvalues in the range of $[0, 2]$; we scale it such that the eigenvalues are in the range of $[0, 1]$). It is worth emphasising that here we use notations with the tilde (e.g., $\tilde{G}$, $\tilde{n}$ and $\tilde{m}$) to make the distinction that this graph is, in general, different from, and is typically a subgraph of, the overall graph $G$ discussed at the start of this section, which might be too large or not be fully available at the start of the optimisation;

---

**Algorithm 1** Bayesian Optimisation on Graphs (*BayesOptG*)

1: **Inputs**: Number of random points at initialisation/restart $N_0$, total number of iterations $T$, subgraph size $Q$, graph $G = \{\mathcal{V}, \mathcal{E}\}$ (whose topology is not necessarily fully known a-priori).
2: **Objective**: The node $v_T^*$ that minimises the objective function $f(v), v \in \mathcal{V}$.
3: **Initialise** `restart_flag` $\leftarrow$ `True`, visited nodes $\mathcal{S} \leftarrow \emptyset$, train data $\mathcal{D}_0 = \emptyset$, $h \leftarrow 1$.
4: **for** $t = 1, ..., T$ **do**
5:     **if** `restart_flag` **then**
6:         Initialise the GP surrogate $\mathcal{D}_t$ with randomly selected $N_0$ points from $\mathcal{V}_{\setminus \mathcal{S}}$ and their observations $\mathcal{D}_t \leftarrow \{v_i, y_i\}_{i=1}^{N_0}$.
7:     **end if**
8:     Construct subgraph $\tilde{G}_t = \{\tilde{\mathcal{V}}_t, \tilde{\mathcal{E}}_t\}$ around $v_t^*$ (best node seen *from the last restart*) (See Algorithm 2 & §3.2).
9:     Fit a GP with kernel defined in Table 1 on $\tilde{G}_t$ with $\mathcal{D}_t$ by optimising log-marginal likelihood.
10:    Select next query point $v_{t+1}$ by optimising the acquisition function.
11:    Query objective function $f(\cdot)$ at $v_{t+1}$ to obtain a (potentially noisy) estimate $y_{t+1}$; update train data $\mathcal{D}_{t+1} \leftarrow \mathcal{D}_t \cup (v_{t+1}, y_{t+1})$; seen nodes $\mathcal{S} \leftarrow \mathcal{S} \cup v_{t+1}$; determine the state of `restart_flag` with the criteria described in §3.2.
12: **end for**
13: **return** node that minimises $f(\cdot)$ *from all restarts*.

---

we defer a full discussion on this in §3.2. We further note that $\tilde{\mathbf{L}} = \mathbf{U}\boldsymbol{\Lambda}\mathbf{U}^\top$ with $\boldsymbol{\Lambda} = \mathrm{diag}(\lambda_1, ..., \lambda_{\tilde{n}})$ and $\mathbf{U} = [\mathbf{u}_1, ..., \mathbf{u}_{\tilde{n}}]$, where $\{\lambda_1, ..., \lambda_{\tilde{n}}\}$ are the eigenvalues of $\boldsymbol{\Lambda}$ sorted in an ascending order and $\{\mathbf{u}_1, ..., \mathbf{u}_{\tilde{n}}\}$ are the corresponding (unit) eigenvectors.

Let $p, q \in \{1, ..., \tilde{n}\}$ be two indices over the nodes of $\tilde{G}$, we may express our covariance function to compute the covariance between an arbitrary pair of nodes $v_p, v_q$ in terms of a *regularisation function* of eigenvalues $r(\lambda_i) \, \forall i \in \{1, ..., \tilde{n}\}$, as described in Smola & Kondor [37]:

$$k(v_p, v_q) = \sum_{i=1}^{\tilde{n}} r^{-1}(\lambda_i) u_i[p] u_i[q], \tag{1}$$

where $u_i[p]$ and $u_i[q]$ are the $p$-th and $q$-th elements of the $i$-th eigenvector $\mathbf{u}_i$. The specific functional form of $r(\lambda_i)$ depends on the kernel choice, and the kernels considered in this work are listed in Table 1. We note that all kernels encode the smoothness of the function on the local subgraph $\tilde{G}$. In particular, the diffusion kernel has been adopted in Oh et al. [27]; the polynomial and Matérn kernels are inspired by recent work in the literature of graph signal processing [11, 46, 3]; finally, the sum-of-inverse polynomials kernel is designed as a variant of the polynomial kernel: in terms of the regularisation function, it can be interpreted as (while ignoring $\epsilon$) a scaled harmonic mean of the different degree components of the polynomial kernel. We next discuss the behaviours of these kernels from the perspective of kernel hyperparameters.

**Kernel hyperparameters.** $\boldsymbol{\beta} := [\beta_0, ..., \beta_{\eta-1}]^\top \in \mathbb{R}_{\geq 0}^\eta$ (for polynomial and sum-of-inverse polynomials) or $[\beta_1, ..., \beta_{\tilde{n}}]^\top \in \mathbb{R}_{\geq 0}^{\tilde{n}}$ (for the diffusion kernel) define the characteristics of the kernel. We constrain $\boldsymbol{\beta}$ in both kernels to be non-negative to ensure the positive semi-definiteness of the resulting covariance matrix and are learned jointly via GP log-marginal likelihood optimisation. The parameter $\nu$ controls the mean-square differentiability in the classical GP literature with the Matérn kernel. The polynomial and the sum-of-inverse polynomials kernels in Table 1 feature an additional hyperparameter of *kernel order* $\eta \in \mathbb{Z}_{\geq 0}$. We set it to be $\min\{5, \textit{diameter}\}$ where *diameter* is the

Table 1: Kernels considered in terms of the regularisation function $r(\lambda_i)$. We derive the semi-definiteness of polynomial and sum-of-inverse polynomial kernels in App. A.

| Kernel | Regularisation function $r(\lambda_i)$ | Kernel function $K(\mathcal{V}, \mathcal{V})$ |
|---|---|---|
| Diffusion[†] [37, 27] | $\exp(\beta_i \lambda_i)$ | $\sum_{i=1}^{\tilde{n}} \exp(-\beta_i \lambda_i) \mathbf{u}_i \mathbf{u}_i^\top$ |
| Polynomial* | $\sum_{\alpha=0}^{\eta-1} \beta_\alpha \lambda_i^\alpha + \epsilon$ | $\sum_{i=1}^{\tilde{n}} \left( \sum_{\alpha=0}^{\eta-1} \beta_\alpha \lambda_i^\alpha + \epsilon \right)^{-1} \mathbf{u}_i \mathbf{u}_i^\top$ |
| Sum-of-inverse polynomials* | $\left( \sum_{\alpha=0}^{\eta-1} \frac{1}{\beta_\alpha \lambda_i^\alpha + \epsilon} \right)^{-1}$ | $\sum_{i=1}^{\tilde{n}} \left( \sum_{\alpha=0}^{\eta-1} \frac{1}{\beta_\alpha \lambda_i^\alpha + \epsilon} \right) \mathbf{u}_i \mathbf{u}_i^\top$ |
| Matérn [4] | $\left( \beta\nu + \lambda_i \right)^\nu$ | $\sum_{i=1}^{\tilde{n}} \left( \beta\nu + \lambda_i \right)^{-\nu} \mathbf{u}_i \mathbf{u}_i^\top$ |

[†]Can be ARD or non-ARD: for ARD, $\{\beta_i\}_{i=1}^{\tilde{n}}$ coefficients are learned; for non-ARD, a single, scalar $\beta$ is learned.

*$\{\beta_\alpha\}_{\alpha=0}^{\eta-1}$ coefficients to be learned. $\epsilon$: small positive constant (e.g. $10^{-8}$). $\eta$: order of kernel.

length of the shortest path between the most distanced pair of nodes in $\tilde{G}$ (a thorough ablation study on $\eta$ is presented in App. D.). We argue that this allows both kernels to strike a balance between expressiveness, as all eigenvalues contained in the graphs are used in full without truncation, and regularity, as fewer kernel hyperparameters need to be learned. This is in contrast to, for example, diffusion kernels on graphs in Table 1, which typically has to learn $\tilde{n}$ hyperparameters for a graph of size $\tilde{n}$, whose optimisation can be prone to overfitting. To address this issue, previous works often had to resort to strong sparsity priors (e.g. horseshoe priors [6]) and approximately marginalising with Monte Carlo samplers that significantly increase the computational costs and reduce the scalability of the algorithm [27]. In contrast, by constraining the order of the polynomials to a smaller value, the resulting kernels may adapt to the behaviour of the target function and can be better regularised against overfitting in certain problems, as we will validate in §5.

## 3.2 Tractable Optimisation via Local Modelling

As discussed previously, it is a technical challenge to develop high-performing yet efficient methods in 1) large, real-world graphs (e.g. social network graphs) and 2) graphs for which it is expensive, or even impossible, to obtain complete topological information beforehand (e.g. if we model the interactions between individuals as a graph, the complete topology of the graph may only be obtained after exhaustive interviews and contact tracing with all people involved). The previous work in Oh et al. [27] cannot handle the second scenario and only addresses the first issue by assuming a certain structure of the graph (e.g. the Cartesian product of subgraphs), but these techniques are not applicable when we are dealing with a general graph $G$.

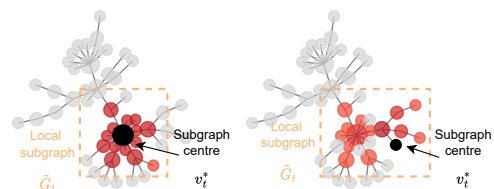

Figure 2: Subgraphs $\tilde{G}_t$ determined by Algorithm 2, marked in red, with a darker shade denoting a closer distance to the central node $v_t^* = \arg\min_{v \in \{v_t\}_{t'=1}^t} f(v)$ in the figure, marked in black), for a high-degree node (**Left**) and a node far from high-degree node (**Right**). Note for the latter case, the local subgraph can include nodes that are much further away.

To address the dual challenges, and inspired by trust region-based BO methods [7, 13, 43, 10, 44], we adapt and simplify the techniques to our use case: we propose to leverage *local modelling* by focusing on a subset of nodes that evolves as the optimisation progresses. At iteration $t \in \{1, ..., T\}$, assuming the collection of our observed configurations and outputs is $\mathcal{D}_t = \{v_{t'}, y_{t'}\}_{t'=1}^t$, we first find the node that leads to the best objective function so far $v_t^* = \arg\min_{v \in \{v_t\}_{t'=1}^t} f(v)$. We then use Algorithm 2 to select a *neighbourhood* around $v_t^*$ that is a subgraph of the overall graph $G$: $\tilde{G}_t \subseteq G$ with $Q$ number of nodes (we will discuss how to choose $Q$ in the next paragraph), in a procedure similar to the neighbourhood sampling in the GraphSAGE framework [18] as illustrated in Fig. 2: in particular, during sampling, the closer nodes to $v_t^*$ takes precedence over further nodes – we only sample the latter if the subgraph consisting of $v_t^*$ and the closer nodes has fewer than $Q$ nodes; hence the local subgraph is a form of an *ego-network* of the central node $v_t^*$. We then only impose the GP and compute the covariance matrix *over this subgraph only*: First, this effectively limits the computational cost – note that the time complexity in our case depends on *both* the number of training examples $N$ and the size of the graph $\tilde{n}$ we impose the GP on ($\mathcal{O}(\tilde{n}^3 + N^3)$), assuming a naïve eigen-decomposition algorithm. Second, it also effectively addresses the setup where the

entire $G$ is not available a-priori, as we only need to query and reveal the topological structure of the subgraph $\tilde{G}_t$ *on the fly*.

---

**Algorithm 2** Selecting a local subgraph

---

1: **Inputs**: Best input up to iteration $t$ *since the last restart*: $v_t^*$, subgraph size $Q$.
2: **Output**: local subgraph $\tilde{G}_t$ with $Q$ nodes.
3: Initialise: $\tilde{\mathcal{V}}_t \leftarrow \{v_t^*\}$, $h \leftarrow 1$.
4: **while** $|\tilde{\mathcal{V}}_t| < Q$ **do**
5:     Find $\mathcal{N}_h$, the $h$-hop neighbours of $v_t^*$.
6:     **if** $|\tilde{\mathcal{V}}_t| + |\mathcal{N}_h| \leq Q$ **then**
7:         Add all $h$-hop neighbours to $\tilde{\mathcal{V}}_t$: $\tilde{\mathcal{V}}_t \leftarrow \tilde{\mathcal{V}}_t \cup \mathcal{N}_h$.
8:         Increment $h$: $h \leftarrow h + 1$
9:     **else**
10:        Randomly sample $Q - |\tilde{\mathcal{V}}|_t$ nodes from $\mathcal{N}_h$ and add to $\tilde{\mathcal{V}}_t$
11:     **end if**
12: **end while**
13: **return** the subgraph $\tilde{G}_t$ induced by $\tilde{\mathcal{V}}_t$ (i.e. the *ego-network*).

---

**Determining the local subgraph size.** The local subgraph size at iteration $t$ ($Q_t$) is a hyperparameter of the algorithm. We adapt the idea of *trust regions* from trust region-based optimisation algorithms [7, 13, 43] to adaptively set the size of $Q_t$ as the optimisation progresses: specifically, we initialise $Q_0$ (initial neighbourhood size), `succ_tol` (success tolerance), `fail_tol` (failure tolerance) and $\gamma > 1$ (multiplier) as hyperparameters[3], and count "successes" as occasions where *BayesOptG* succeeds in improving the function values (i.e., at iteration $t$, $f(v_t^*) < f(v_{t-1}^*)$) and "failures" otherwise. Upon consecutive `succ_tol` successes, we expand the neighbourhood size $Q_t \leftarrow \min(\text{round}(\gamma Q_{t-1}), n)$ to increase exploration, and upon consecutive `fail_tol` failures, we shrink $Q_T \leftarrow \max(\text{round}(Q_{t-1}/\gamma, Q_{\min}))$ to increase exploitation. The notation $\text{round}(\cdot)$ denotes rounding to the nearest integer, and $Q_{\min}$ denotes some minimum value of neighbourhood size (typically set to 1 to include a single node $v_t^*$ for simplicity, although alternative values may be used). When $Q_t \leq Q_{\min}$, we *restart* the BO by fitting the surrogate with randomly initialised nodes whose objective function values have not been evaluated.

**Remarks on the relation to trust-region BO methods.** It is worth noting that while conceptually influenced by previous trust region-BO methods, the local graph construction we use differs from these methods in several crucial aspects. First, we use a bespoke distance metric in the graph space. Second, whereas the purpose of trust regions in previous works is to alleviate over-exploration in high-dimensional spaces, local subgraphs in our case also uniquely serve the crucial purpose of allowing *BayesOptG* to handle imperfect knowledge about the graphs, as we only need to reveal the topology of the subgraph (as opposed to the entire graph) at any given iteration. Lastly, we discussed, that using trust regions also improves scalability – this can be concretely exemplified by the massive speed-up shown in Fig. 3.

**Optimisation of the acquisition function.** With the local subgraph obtained, we then fit a GP surrogate with the covariance function defined in §3.1 and optimise log-marginal likelihood. Given that the local search space in our case is finite (of size $Q$), we simply enumerate all nodes *within* $\tilde{G}_t$ to compute their *acquisition function* $\text{acq}(\cdot)$ values (which is computed from the predictive mean and variance of the GP surrogate) and pick the maximiser as the recommended location to query the *objective function* $f(\cdot)$ for iteration $t + 1$ as

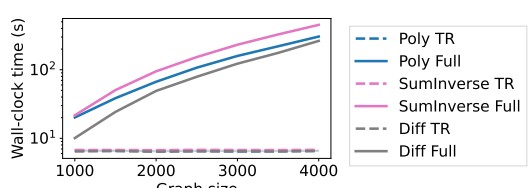

Figure 3: *Trust regions enable efficient optimisation on large graphs*: Wall-clock time with and without trust regions in *BayesOptG* with different kernels over graphs of different sizes.

$v_{t+1} = \arg\max_{v \in \tilde{\mathcal{V}}_t} \text{acq}(v)$. Any off-the-shelf acquisition function may be used, and we adopt expected improvement (EI) [16] in our experiments. It is worth noting that *BayesOptG* is also fully compatible with existing approaches such as Kriging believer fantasisation [17] for batch BO.

# 4 Related Work

The setup we consider is by itself novel and largely under-explored. One of the few existing methods that can be used for optimisation over a graph search space is COMBO [27], where the search space is modelled as a graph that captures the relationship between different values for a group of

---

[3] We provide an ablation study in App. D to show the robustness of *BayesOptG* to hyperparameters.

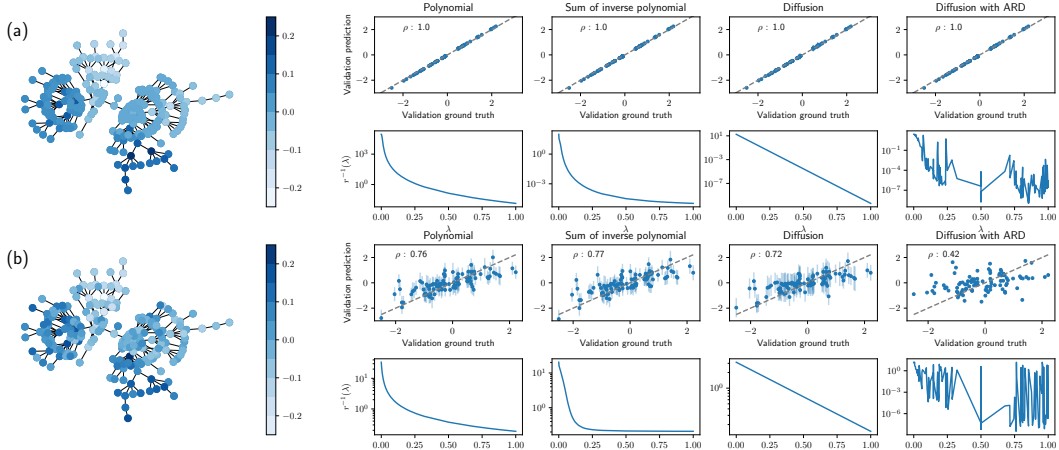

Figure 4: Validation of predictive powers of kernels considered on a BA graph of size $n = 200$ nodes and parameter $m = 1$, with **(a)** function values on the nodes corresponding to elements of the eigenvector corresponding to the second smallest eigenvalue and **(b)** same as above, but corrupted with noise standard deviation $\sigma = 0.05$. The leftmost column shows the visualisation of the ground truth, and the right columns show the GP posterior mean and standard deviation (error bars) learned by the different kernels against ground truth with Spearman correlation $\rho$ and learned $r^{-1}(\lambda)$ (Eq. 1).

categorical variables. It is, therefore, designed explicitly for combinatorial optimisation. Several studies modified COMBO in various ways but followed essentially the same framework for similar tasks, e.g., optimisation over categorical variables [12, 19, 24]. Similarly, Ramachandram et al. [30] propose a specific graph construction to optimise multimodal fusion architectures. Our work differs from these studies in that: 1) we focus on optimisation over generic, large-scale and potentially unknown graphs; 2) the nodes of the graph are not limited to combinations of values for categorical variables and can represent any entities; 3) the kernel we propose is not limited to diffusion-based ones and can adapt to the behaviour of the function to be optimised. Finally, the *graph bandit setting* ([5, 39, 38]) can be seen to be similar to ours in the sense that it also aims at finding extreme values associated with nodes in a graph. However, the bandit problem considers a stochastic setting where nodes are influenced in a probabilistic fashion, and the objective function is actively shaped by this process; in comparison, in our case, we consider an underlying deterministic and black-box function, which is more aligned with the classical BO setting. Moreover, both Valko et al. [39] and Thaker et al. [38] require *full* graph access and require prohibitive operation on the full graph Laplacian (decomposition/inversion), whereas *BayesOptG* may work on-the-fly with initially unknown graphs and is much more scalable thanks to the designs in §3.2. Several works also leverage kernels on graphs to build Gaussian processes for graph-structured data [26, 41, 40, 46, 28, 4, 29]. While the kernels proposed in these approaches can, in theory, be used in a BO framework, these studies do not address the optimisation problem we consider.

Another line of work focuses on optimisation *over graph inputs* (in contrast to a *graph search space*) where each input configuration itself is a graph. In contrast, in our case, each input configuration is a *node*. Examples of the former include Ru et al. [34] who model neural architectures as graphs and use Weisfeiler-Lehman kernels [36] to perform BO, and Wan et al. [42], who devise a BO agent for adversarial attack on graph classification models. Other representative examples include Kandasamy et al. [20], Korovina et al. [23] and Cui et al. [8, 9]. We emphasise that, while related, these works deal with a different setup and thus require a different method compared to the present work. For example, the kernels over graphs used in these methods typically aim to find vector embedding of graphs that account for their topologies. However, once the embedding is computed, standard Euclidean covariance functions (e.g., the dot product or squared-exponential kernel) are applied. On the other hand, in the present work, we aim to compute similarities over nodes, where topological information is crucial *during* the covariance computation itself.

## 5   Experiments

We first validate the predictive power of the GPs with the adopted kernels on graphs and then demonstrate the optimisation performance of *BayesOptG* in both synthetic and real-world tasks. We compare *BayesOptG* against baselines, including random and local search optimisation algorithms as

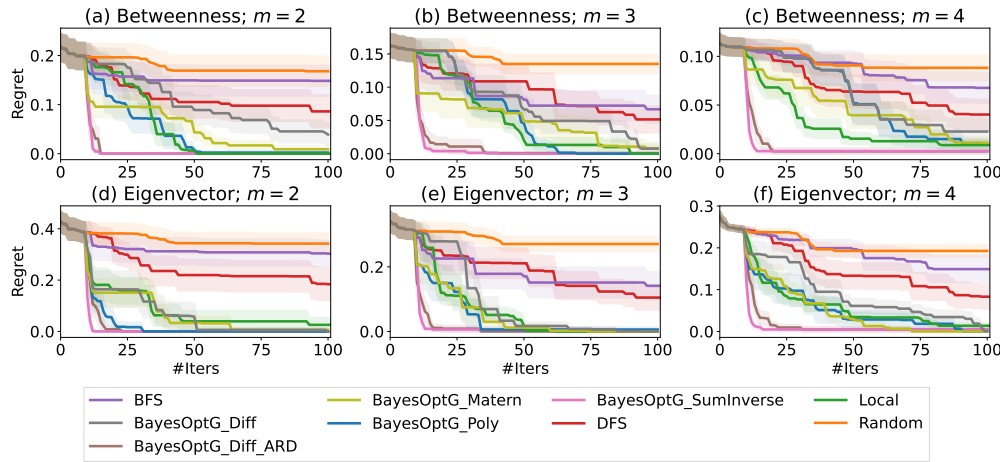

Figure 5: *Maximising centrality scores* with the **BA** random graph model and $n = 1000$ nodes. Different graphs show different values of the BA hyperparameter $m \in \{2, 3, 4\}$ and centrality metrics {betweenness/eigenvector centrality}.

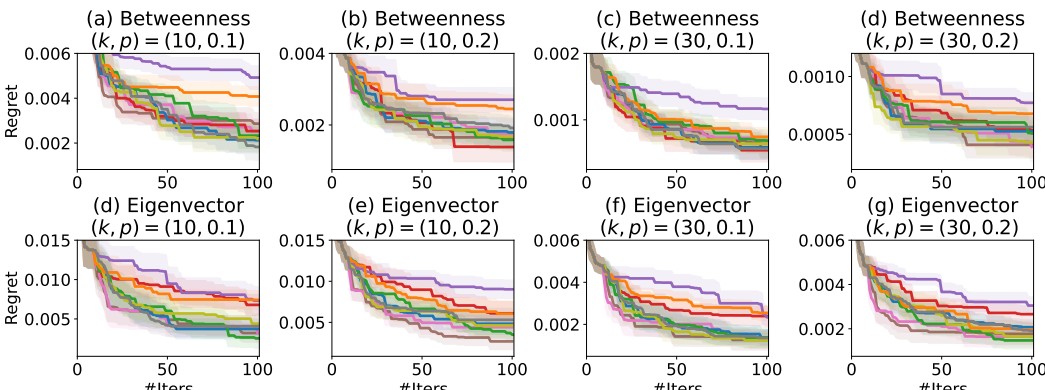

Figure 6: *Maximising centrality scores* with the **WS** random graph model and $n = 2000$ nodes. Refer to Fig. 5 for legend and additional explanations.

well as BFS and DFS. The description of these baselines is given in the App. B.2. In all figures, lines, and shades denote mean and standard error, respectively, across ten trials.

## 5.1 Validating Predictive Power of Kernels

We first validate the predictive power of the adopted kernels in controlled regression experiments. To do so, we generate functions that are simply the eigenvectors of the graph Laplacian and compare the predictive performance of the kernels using three graph types: 2D grid, Barabási–Albert (BA) [1] and Watts–Strogatz (WS) [45]. We compare the performance in terms of validation error and show the results in Fig. 4 (results for other graph types are shown in App. C.1). We find that in the noiseless case, all kernels learn the underlying function effectively (except that the diffusion with ARD kernel learns a non-smooth transform on the spectrum due to its over-parameterisation, resulting in underestimations of the uncertainty in the noisy case). Still, the better-regularised kernels (described in §3.1) are considerably more robust to noise corruption.

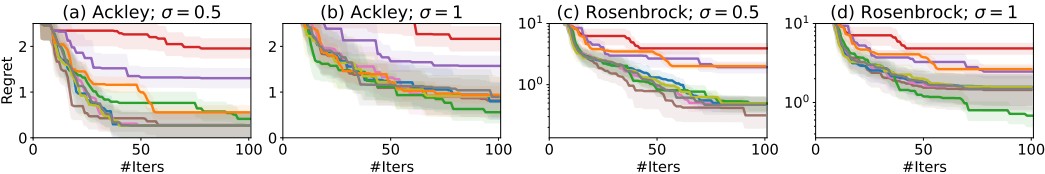

Figure 7: *Synthetic test functions* task with `Ackley`/`Rosenbrock` functions with noise standard deviation $\sigma \in \{0.5, 1\}$. Regrets shown in log-scale for `Rosenbrock`; refer to Fig. 5 for legend.

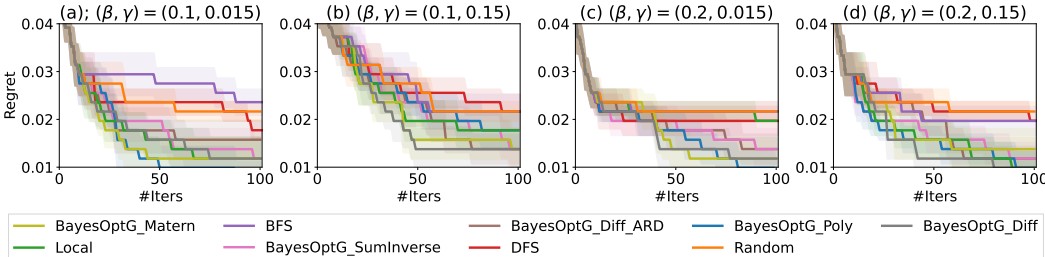

Figure 8: *Identifying the patient zero* task with different SIR model hyperparameters $\beta \in \{0.1, 0.2\}$ and $\gamma \in \{0.015, 0.15\}$ and probability of recovery $\epsilon$ of 0. Refer to Fig. 20 – 23 for experiments with other hyperparameter combinations.

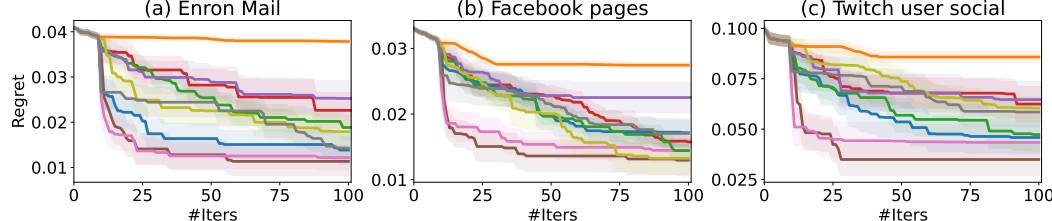

Figure 9: *Identifying influential users in a social network* task on different real-life social networks (Enron/Facebook page/Twitch). Refer to Fig. 8 for legend.

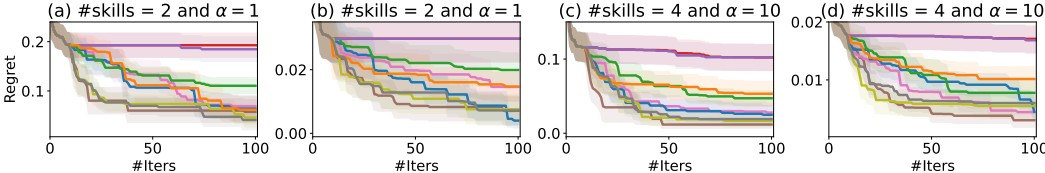

Figure 10: *Team optimisation* task with $s$ (number of skills) $\in \{2, 4\}$ and $\alpha \in \{1, 10\}$ with Jaccard index threshold of 0.3 (refer to App. B.4.3 for explanations). Refer to Fig. 8 for legend and Fig. 24 – 26 for experiments with other hyperparameter combinations.

## 5.2 Optimisation Tasks

We conduct experiments on a number of synthetic and real-life tasks that involve or imitate expensive optimisation, and we show all results in terms of *simple regret* (i.e., the difference between the objective function value and the ground-truth optimum). We consider the following synthetic tasks:

- **Maximising centrality scores** (Fig. 5 and 6; Fig. 18 in App. C.2): we aim to find the node with maximum centrality measure, from a graph sampled from a random graph model. We consider both *eigenvector centrality* and *betweenness centrality* as the centrality metrics, and use BA and WS with different hyperparameters as the random graph-generating models. We consider graphs with sizes in the range of $10^3$ in Fig. 5 and 6. In Fig. 18, we further scale the size of graphs considered to $10^6$ nodes to demonstrate the scalability of our method in a large-scale setup.

- **Synthetic test functions** (Fig. 7): we optimise a suite of discretised versions of commonly used synthetic test functions (`Ackley` and `Rosenbrock`) on graphs defined as a 2D-grid in both noiseless and noisy setups. The readers are referred to App. B.3.2 for additional implementation details.

We consider the following real-life tasks:

- **Identifying the patient zero** (Fig. 8; Fig. 20 to 23 in App. C.3): we aim to find the "patient zero" of an epidemic in a contact network, who is to the person identified as the first carrier of a communicable disease in an epidemic outbreak. We use a real-world contact network based on Bluetooth proximity [2], and on top of simulating the epidemic process using the *SIR model*, the canonical compartmental model in epidemiology [22]. The function values are the time instants when an individual is infected; the readers are referred to App. B.4.1 for more details of this task.

- **Identifying influential users in a social network** (Fig. 9): we aim to find the most influential user in a social network. There are multiple ways of defining the influence power of a user, and for simplicity, we follow the common practice of taking *node degree* as a proxy of influence [21]. We

use three real-world networks, namely the *Enron email network* [25], *Facebook page network* [33], and *Twitch social network* [32]. The readers are referred to App. B.4.2 for more details.

- **Team optimisation** (Fig. 10; Fig. 24 to 26 in App. C.4): we design a task of optimising team structure, where the objective is to find a team that contains members who are experts in different skills, and their collective expertise represents a diverse skill set. In this case, the teams are modelled as nodes, and edges represent the a priori similarity between teams. While there are various possible ways to model these similarities, in our experiment, we consider that an edge exists between two nodes if the Jaccard index between the two sets of team members is greater than a certain threshold. We include additional details and a formal description of the objective function in the App. B.4.3.

We designed these tasks to imitate expensive but realistic black-box optimisation problems on which the use of Bayesian optimisation is ideal. For example, the *identifying patient zero* task imitates real-life contact tracing. If executed in real life, each function evaluation requires expensive and potentially disruptive procedures like interviews about the individuals' travel history and the people they were in contact with. On the other hand, the *centrality maximisation & identifying influential social network users* problems mirror common online advertising tasks to identify the influential users without access to the full social network information (which would be near-impossible to obtain given the number of users). Real-life social media often limits how much one may interact with their platform through pay-per-use APIs or hard limits (e.g. upper limit of views). In either case, there is a strong reason to identify the influential users in the most query-efficient manner.

**Discussions.** In addition to the task-specific results, we further aggregate the performance of the different methods over all tasks in terms of relative ranking in Fig. 11. We find that within individual tasks and aggregated across the different tasks, *BayesOptG with any kernel choice* generally outperforms all baselines in terms of efficiency, final converged values, or both. Specifically, *Random* is simple but typically weak for larger graphs, except for very rough/noisy functions (like `Ackley`), or the variation in function values is generally small; *DFS* and *BFS* are relatively weak as they consider graph topology information only but not the node information (on which the objective function is defined) and can be sensitive to initialisation; *Local search* is, on balance, the strongest baseline, and it does particularly well on smoother functions with fewer local minima.

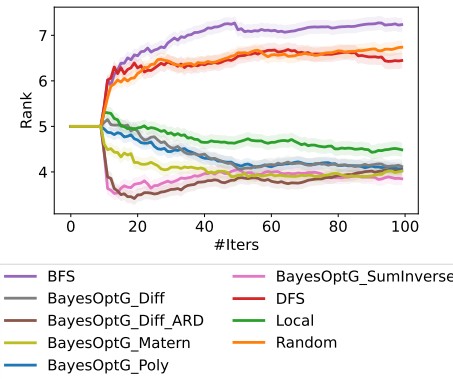

Figure 11: Aggregated ranks of the methods (lower is better) vs. the number of evaluations averaged across all experiments.

As is the case for any GP-based method, the kernel choice impacts the performance, and the performance is stronger when the underlying assumptions of the kernel match the actual objective function. For example, diffusion kernels work well for patient zero identification (Fig. 8) and team optimisation (Fig. 10), as the underlying generative functions for both problems, are indeed smooth (in fact, the SIR model in disease propagation is heavily connected to diffusion processes). Diffusion without ARD further enforces isotropy, assuming the diffusion coefficient in all directions is the same, and thus typically underperforms except for team optimisation, where the generated graph is well structured and `Ackley`, which is indeed isotropic and symmetric. We recommend only if we know that the underlying function satisfies its rather stringent assumptions. Finally, the SumInverse and DiffARD kernels are generally better, as they offer more flexibility in learning from the data; *we recommend using one of these as default without prior knowledge suggesting otherwise*.

## 6 Conclusion

We address the problem of optimising over functions on graphs, a hitherto under-investigated problem. We demonstrate that BO, combined with learned kernels on graphs and efficient local modelling, provides an effective solution. The proposed framework works with generic, large-scale and potentially unknown graphs, a setting that existing BO methods cannot handle. Results on a diverse range of tasks support the effectiveness of the proposed method. The current work, however, only considers the case where the optimisation is over *nodes*; possible future works include extensions to related settings, such as optimising over functions defined on *edges* and/or on hypergraphs.

# Acknowledgement

The authors would like to acknowledge the following sources of funding in direct support of this work: X.W. is supported by the Clarendon Scholarship at University of Oxford; P.O. is supported by the EPSRC Centre for Doctoral Training in Autonomous Intelligent Machines and Systems EP/L015897/1; X.D. acknowledges support from the Oxford-Man Institute of Quantitative Finance and the EPSRC (EP/T023333/1). The authors declare no conflict of interest.

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

# A  Proof of Semi-Definiteness

In this section, we show that all kernels considered in this paper are positive semi-definite (p.s.d). Specifically, we note that using the terminology defined in Eq. 1, any map $r \to \mathbb{R} \to [0, +\infty]$ defines a valid covariance kernel. Indeed,

$$\forall \mathbf{X} \subset \mathcal{V}, k(\mathbf{X}, \mathbf{X}) = \sum_{i=1}^{\tilde{n}} r^{-1}(\lambda_i) \mathbf{u}_i[\mathbf{X}] \mathbf{u}_i[\mathbf{X}]^\top, \tag{2}$$

where $\mathbf{u}_i[\mathbf{X}] = \left[ u_i[x_1], u_i[x_2], ..., u_i[x_l] \right]^\top$ with $l = |\mathbf{X}|$. The matrix $u_i[\mathbf{X}]u_i[\mathbf{X}]^\top$ is symmetric p.s.d as the outer product of one non-zero vector: $\forall \mathbf{x} \in \mathbb{R}^l, \mathbf{x}^\top u_i[\mathbf{X}]u_i[\mathbf{X}]^\top \mathbf{x} = \|u_i[\mathbf{X}]^\top x\|_2^2 \geq 0$. As a result, our covariance matrix is symmetric p.s.d as the weighted sum of symmetric positive semidefinite matrices with positive coefficients. The kernels we presented in this paper correspond to a positive $r$; hence, they are all p.s.d.

# B  Experimental Details

## B.1  Random Graph Models

**Barabási–Albert model (BA).** The network begins with an initial connected network of $m_0$ nodes. New nodes are added to the network one at a time. Each new node is connected to $m \leq m_0$ existing nodes with a probability that is proportional to the number of links that the existing nodes already have. The probability $p_i$ that the new node is connected to node $i$ is:

$$p_i = \frac{k_i}{\sum_j k_j}$$

where $k_j$ is the degree of node $j$.

**Watts–Strogatz model (WS).** The WS model was introduced to explain the "small-world" phenomena in a variety of networks. It achieves this by interpolating between a randomized structure close to ER graphs and a regular ring lattice. Given a mean degree $K$ and a parameter $\beta \in [0, 1]$. An undirected graph is constructed with $N$ nodes and $NK/2$ edges as follows:

1. Constructs a regular one-dimensional network with only local connections of range $K$, meaning each node is connected to its $K/2$ nearest neighbours on each side.

2. For every node $i = 0, ..., N-1$ take every edge connecting $i$ to its $K/2$ rightmost neighbours. Rewire each of these edges with probability $\beta$ to random nodes while avoiding self-loop and link duplicates.

## B.2  Baseline Algorithms

**Breadth first search (BFS) and depth-first search (DFS).** These algorithms aim to explore the whole graph data structure. It starts with a root node and explores according to the depth or breadth of the graph. In the former, the algorithm explores as far as possible along each branch before backtracking. In the latter, the algorithm explores all nodes at the present depth prior to moving on to the nodes at the next depth level.

**Random search.** In this algorithm, at each time step, a random node is selected for evaluation of our objective function.

**Local search.** In this algorithm, at each time step, we sample and query a random node from a neighbour of the node of the maximum value encountered so far, and we move to a neighbour node if the queried value is better than the incumbent best. When the algorithm reaches a local optimum (i.e., all neighbours have worse values than the current optimum), we allow our algorithm to restart at a random, unvisited node in the graph.

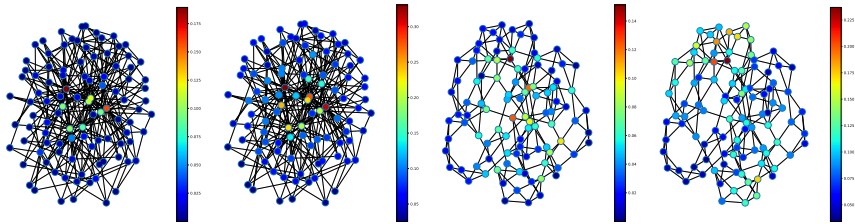

Figure 12: Betweeness/Eigenvector Centrality on BA/WS graphs.

## B.3 Synthetic Optimisation Tasks

### B.3.1 Maximising network centrality

Centrality measures were introduced in network analysis to study the importance of certain vertices with respect to desired characteristics. In this paper, we used two centrality measures: *betweenness* centrality and *eigenvalue* centrality. We show examples of these functions on sample BA/WS graphs in Fig. 12.

**Betweenness centrality.** This centrality focuses not just on overall connectedness but on the occupying positions that are pivotal to the network's connectivity. The following formula gives the betweenness of node $v$:

$$g(v) = \sum_{\substack{s,t \in \mathcal{V} \setminus \{v\} \\ s \neq t}} \frac{\sigma_{st}(v)}{\sigma_{st}},$$

where $\sigma_{st}$ is the total number of shortest paths from node $s$ to node $t$ and $\sigma_{st}(v)$ is the number of those paths that pass through $v$. $\mathcal{V} \setminus \{v\}$ denotes the set of neighbouring nodes of $v$ except the node $v$ itself.

**Eigenvector centrality.** This centrality measure accounts for the influence of a particular node within the network. The centrality score for the whole set of vertices, represented as a vector $\mathbf{x}$, is a solution to the equation:

$$\mathbf{Ax} = \lambda \mathbf{x},$$

where the matrix $\mathbf{A}$ represents the adjacency matrix and $\lambda$ represents the largest eigenvalue of the adjacency matrix.

### B.3.2 Synthetic test functions

In this subsection, we describe the Rosenbrock and Ackley test functions used for our task, both of which are discretised versions of their original, continuous function forms. The mathematical definitions of the test functions are listed below and are visualized in Fig. 13.

**Rosenbrock function**.

$$f(x, y) = 100(y - x^2)^2 + (x - 1)^2$$

**Ackley function**.

$$f(x, y) = -20 \exp\left(-0.2\sqrt{0.5(x^2 + y^2)}\right) - \exp\left(-0.5(\cos 2\pi x + \cos 2\pi y)\right) + 20 + \exp(1)$$

**Additive noise**. In order to alter the smoothness property of our graph signal defined over the grid, we add random noise governed by noise standard deviation $\sigma_n$ to be added to the loss function $\hat{f}(x, y) = f(x, y) + \epsilon$ with $\epsilon \sim \mathcal{N}(0, \sigma_n^2)$. In our experiment, we vary the noise standard deviation $\sigma_n \in \{0, 0.1, 1, 5\}$ for the Ackley function and $\sigma_n \in \{0, 0.1, 0.5, 1\}$ for the Rosenbrock function.

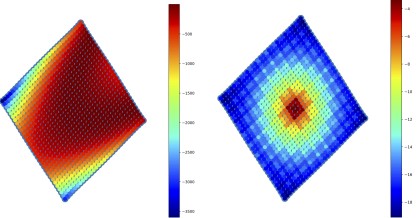

Figure 13: Test function values taken on a regular graph corresponding to the input space: Rosenbrock (**left**); Ackley (**right**).

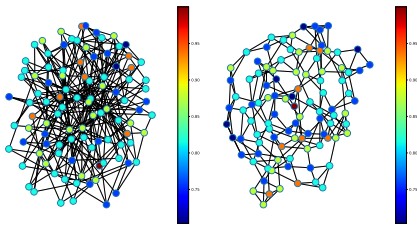

Figure 14: Simulation of the SIR process on BA/WS graphs. It is worth noting that some local optima are visible due to the $\epsilon$ parameter.

### B.4  Real-World Optimisation Tasks

#### B.4.1  Finding patient zero in a contact network

In this task, we simulate the diffusion processes over a graph via epidemics SIR models Kermack & McKendrick [22]. We slightly modify this model to allow for a parameter $\epsilon$ representing the probability of spontaneous infection from unknown factors.

More formally, given a graph $\mathcal{G} = \{\mathcal{V}, \mathcal{E}\}$, our model has three parameters. Parameter $\beta$ encodes the probability of infection, $\beta$ the probability of recovery $\epsilon$ the probability of spontaneous infection, and $T$ the time spent since the beginning of the outbreak. Let time $t = 1, ...T$ be the current time, $\mathbf{x}_{v,t} \in \{I, S, R\}$ the node status (Infected, Susceptible, Recovered) and $\mathcal{S}_{I,t}, \mathcal{S}_{S,t}, \mathcal{S}_{R,t}$ the set of nodes in each category at time t. We have:

$$\forall v \in \mathcal{S}_{I,t}, \begin{cases} \mathbb{P}[\mathbf{x}_{v,t+1} = R] = \gamma \\ \mathbb{P}[\mathbf{x}_{v,t+1} = I] = 1 - \gamma. \end{cases} \tag{3}$$

$$\forall v \in \mathcal{S}_{S,t}, \begin{cases} \mathbb{P}[\mathbf{x}_{v,t+1} = I] = 1 - (1 - \epsilon) \times (1 - \beta)^{|N(v) \cap \mathcal{S}_{I,t}|} \\ \mathbb{P}[\mathbf{x}_{v,t+1} = S] = (1 - \epsilon) \times (1 - \beta)^{|N(v) \cap \mathcal{S}_{I,t}|}. \end{cases} \tag{4}$$

$$\forall v \in \mathcal{S}_{R,t}, \mathbb{P}[\mathbf{x}_{v,t+1} = R] = 1. \tag{5}$$

Given such a process, we construct an objective function indicating how close a certain node is to the source of the infection as follows. At time $T$, where we consider the diffusion to have ended (or corresponding to the present moment when looking for patient zero), for every node in $\mathcal{S}_{R,T} \cup \mathcal{S}_{I,T}$ we denote by $\tau_v$ the first time of infection. The objective function is then defined as:

$$\forall v \in \mathcal{V}, f(v) = \begin{cases} 0 & \text{if } v \in \mathcal{S}_{S,T} \\ (1 - \frac{\tau_v}{T})^2 & \text{if } v \in \mathcal{S}_{I,T} \cup \mathcal{S}_{R,T} \end{cases} \tag{6}$$

This function takes value in $[0, 1]$ and is maximised when the node corresponds to the patient zero. We expect local methods to perform well in this setting as local behaviour can trace the source of the infection through diffusion, and the variation of the functions on the graph is relatively smooth. The introduction of parameter $\epsilon$ nevertheless adds some sources of "local" minima in the graph objective function – we show some examples of such phenomenon in Fig. 14, where we give exemplary graph signals induced by the generative process we described in this section.

### B.4.2 Finding influential users in a social network

In this task, we consider a common problem in identifying the most influential person within a social network. Influence, in this context, is often quantified approximately using degree centrality, which may account for, for example, the number of followers or connections an individual possesses. However, the enormity of social network graphs often restricts our access to complete graph information, necessitating alternative search approaches. To validate our methodology, we conduct tests on various real-world graphs derived from diverse social networks. These include the Enron email network, which represents email communication between members of a corporation; the Facebook page network, which is a network of interconnected Facebook pages; and the Twitch social network, which provides insight into the relationships among users on the Twitch platform.

### B.4.3 Team optimisation

In this task, we aim to tackle the problem of optimising the performance of a team of individuals with different skills. We will assume that a team would perform most effectively when 1) all skills are covered by combining individual skills and 2) some individuals master every skill. More formally, we will consider a pool of $N$ individuals. Each individual can be represented by a vector of skills $\mathbf{x}_i \in [0, 1]^K$ where $K$ is the number of skills. This setup fits our framework well in the scenario where the skills of individuals are unknown, and the pool of potential candidates is also unknown in advance. A sample graph generated from this problem is shown in Fig. 15.

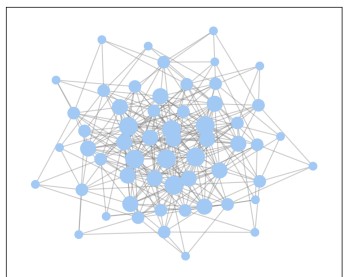

Figure 15: An exemplary graph induced from the team optimisation problem.

**Skill generative process.** In each experiment, we will assume individual skills to be generated according to a Dirichlet distribution with parameter $\alpha$:

$$\mathbf{x}_i \sim \mathrm{Dir}(\boldsymbol{\alpha}), \boldsymbol{\alpha} = [\alpha_1, ..., \alpha_D]^\top$$

The parameter $\alpha$ encodes the sparsity of skill expertise in the general population. Small $\alpha$ generates individuals with specialised skills with more probability than large $\alpha$ where all skill levels concentrate to a score of $0.5$.

**Graph construction.** To solve this task and allow flexible exploration of teams with a varying number of individuals, we construct a graph where nodes represent teams and edges are based on the Jaccard index between each pair of team member sets. More specifically, given two teams $s_1 \subset \mathbb{N}$ and $s_2 \subset \mathbb{N}$ the similarity between them is computed as $w(s_1, s_2) = \frac{s_1 \cap s_2}{s_1 \cup s_2}$. Given $N$ teams, we can then construct an undirected graph with edges:

$$\forall s_1, s_2 \subset [N], (s_1, s_2) \in \mathcal{E} \iff w(s_1, s_2) > \mathrm{Median}(\{w(s_1, s_2) : s_1, s_2 \subset [N]\})$$

**Objective function.** To model the two desirable properties in terms of team composition, we choose the following objective function:

$$\forall s \subset [N] : f(s) = H_k[\mathbb{E}_n[\mathbf{x}]] - \mathbb{E}_n[H_k[\mathbf{x}]]$$

Intuitively, the first term of the objective corresponds to the entropy of the skill distribution of the whole team, which is maximised when the skill distribution is close to the uniform distribution. The second term corresponds to the expected entropy of the distribution of skills of each individual, which is minimised (and the objective maximised) when each individual specialises in one skill. As a result, we can expect this objective to be well suited for modelling an ideal composition of a team.

## C  Additional Experiments

### C.1  Kernel Validation

Complementary to Fig. 4 in the main text, we conduct further regression analyses to confirm the expressive power of the investigated kernels. The results are shown in Fig. 16 and Fig. 17.

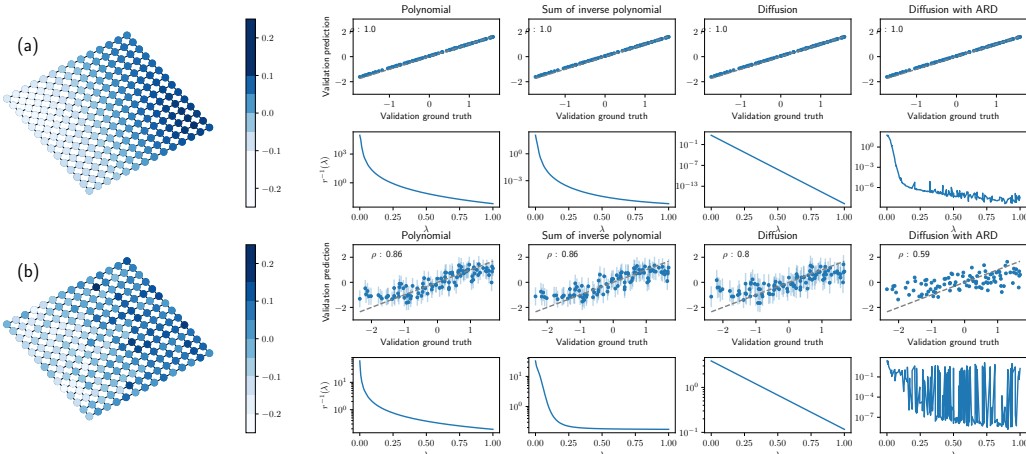

Figure 16: Expressiveness of kernels on a grid graph of size $n = 200$ nodes. Refer to Fig. 4 for more explanations.

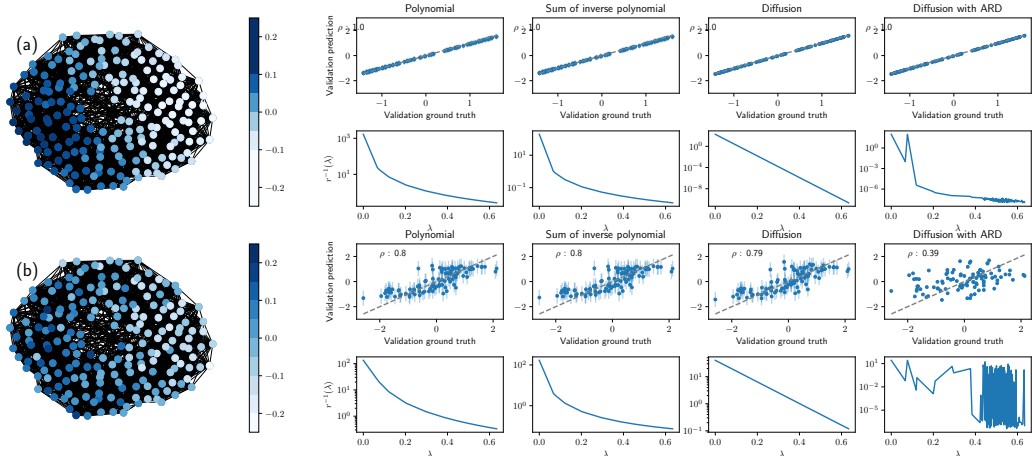

Figure 17: Expressiveness of kernels on a WS graph of size $n = 200$ nodes. Refer to Fig. 4 for more explanations.

## C.2 Centrality Maximisation on Large Graphs

In this section, we consider a similar problem of centrality maximisation as described in App. B.3.1, but on significantly larger graphs: we use BA and WS random graph generators similar to the experiments in Fig. 5 and Fig. 6 in the main text, but we generate graphs with $10^6$ nodes instead and increase the query budget. We show the results in Fig. 18, and we find that the superiority of *BayesOptG* methods persists in this setting over the baseline methods.

## C.3 Finding Patient Zero Task in Real-World Graphs

**Setup.** In this section, we consider several SIR diffusion problems as described in App. B.4.1 where we aim to find the patient zero on a real-world interaction network from the Copenhagen Networks Study [35]. This network represents physical proximity among participants (estimated via Bluetooth signal strength) in a population of more than 700 university students and thus is a good testbed to examine the diffusion process of a hypothetical epidemic outbreak. The visualization of the function on this graph is given by Fig. 19.

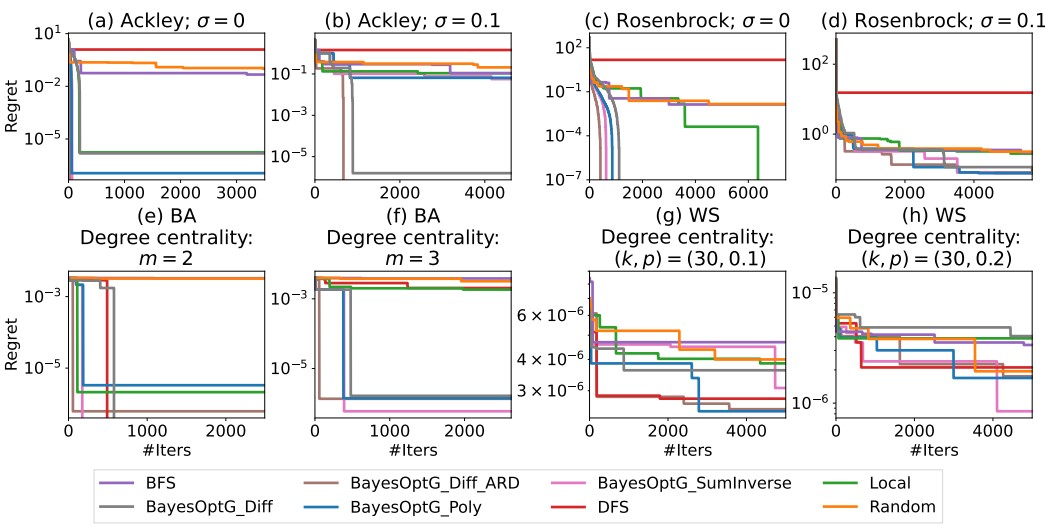

Figure 18: *Maximising centrality scores* with the **BA/WS** random graph model and $n = 10^6$ nodes.

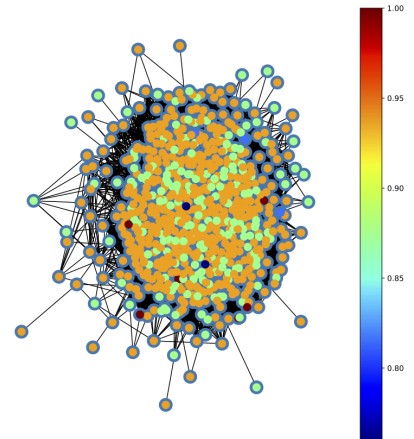

Figure 19: Diffusion objective function on the real-world interaction network.

**Results.** The performance of each algorithm is presented in Fig. 20 − 23 where we use different values of initially infected population fraction and probability of recovery – it is clear that due to the increased complexity as revealed in Fig. 19, there is some performance degradation in all algorithms considered. However, we can see that in most cases, our method performs at least as well as the local search baseline.

## C.4 Team Optimisation

We show additional results for the team optimisation tasks in Fig. 24 to 26. We can observe that the key findings from the main text on this problem (Fig. 10) largely hold true for these tasks induced by different parameters.

# D Ablation and Sensitivity Studies

In this section, we perform a thorough ablation and sensitivity study on how much the additionally introduced hyperparameters affect the algorithm's performance. We report sensitivity analyses to the most important hyperparameters below, namely $Q_0$ (initial trust region size), `fail_tol`, $\eta$

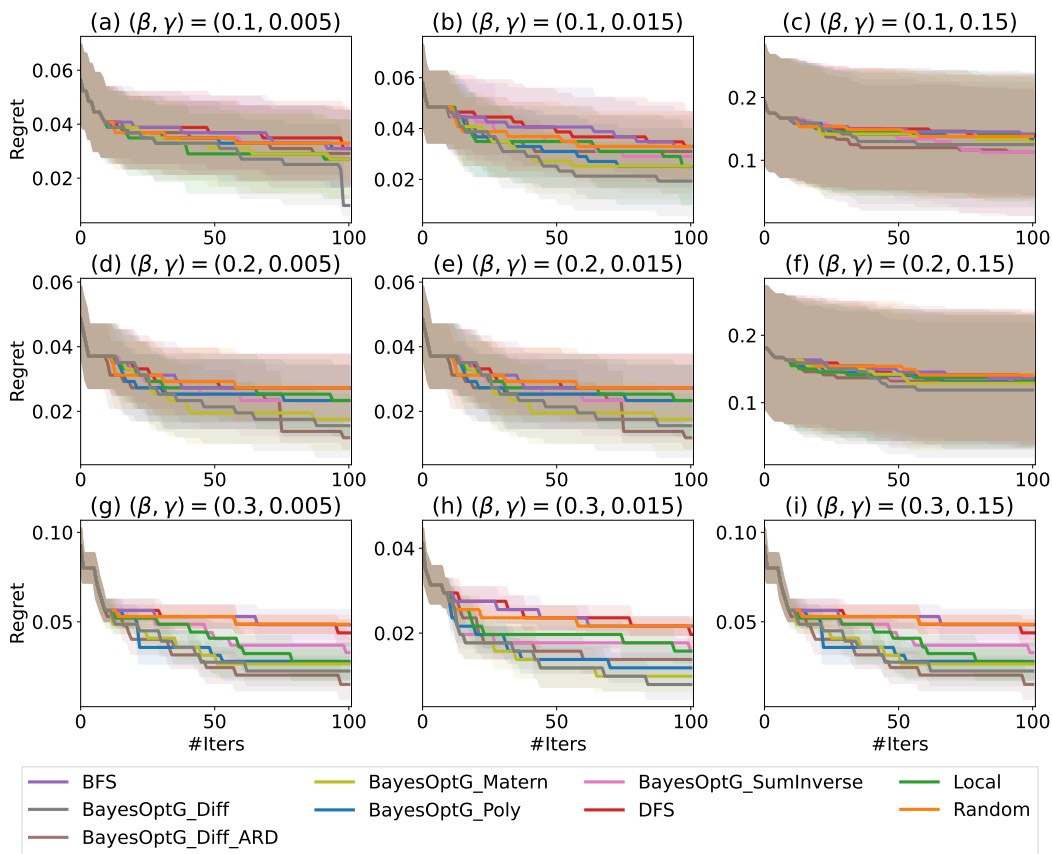

Figure 20: *Identifying the patient zero* task with different SIR model hyperparameters $\beta \in \{0.1, 0.2, 0.3\}$ and $\gamma \in \{0.005, 0.015, 0.15\}$. A fraction of **0.0003** of the initial population was infected initially. The probability of recovery $\epsilon$ is set to **0**.

(order of the kernels) and $\gamma$ (the trust region multiplier in case of successive successes or failures). We also additionally study the effect of introducing the trust region in this section. We perform ablation experiments in the setting with BA graphs and synthetic function optimization. We show the sensitivity analysis in Fig. 27 to 29 – it is evident that our algorithm is largely robust to the choice of hyperparameters as long as a value within a sensible range is chosen.

**Use of trust regions.** We compared, on some relatively small graphs (1000 nodes) in Fig. 31 – as observed, while there is a small drop in performance because of the use of local modelling compared to constructing a surrogate model on the whole graph, it is worth noting that, as shown in Fig. 3 in the main text, the full Bayesian optimisation procedure with kernels defined on the whole graph becomes too prohibitive, even for a relatively small graph of size 1000, and that when the graph is unknown, it is impossible in the first place to construct a whole-graph GP. This verifies that the trust region strikes a promising balance between efficiency and performance.

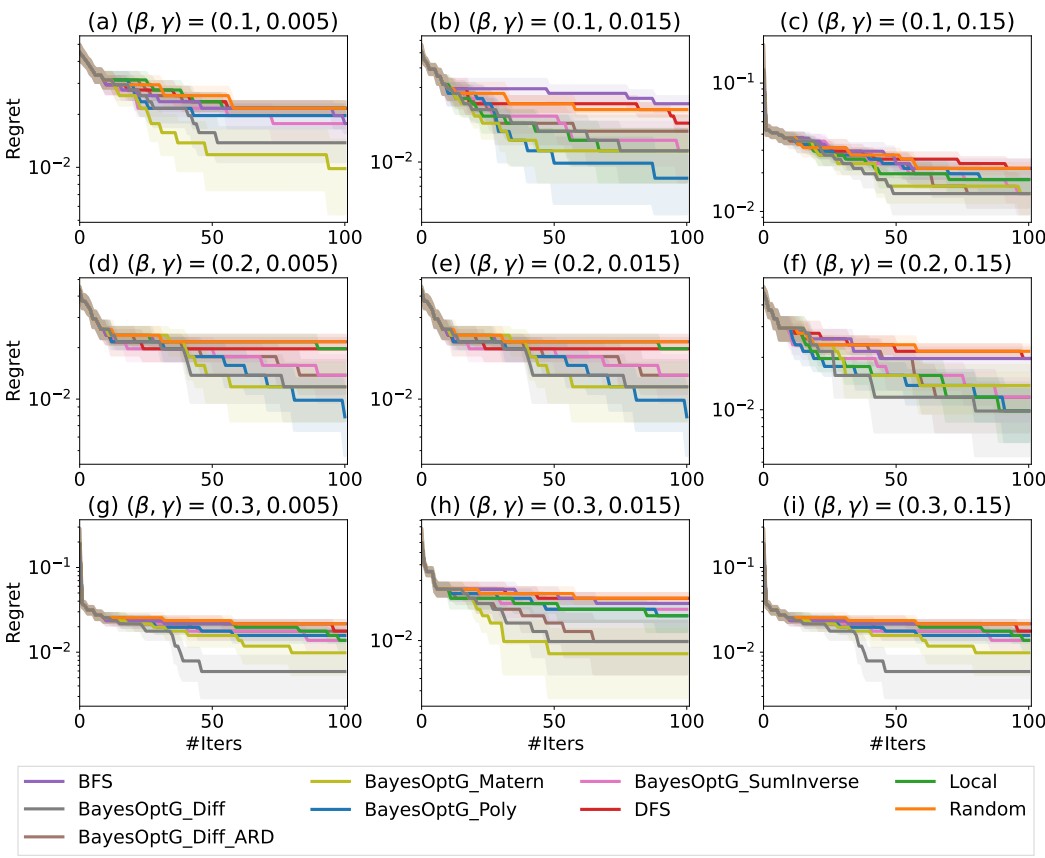

Figure 21: *Identifying the patient zero* task with different SIR model hyperparameters $\beta \in \{0.1, 0.2, 0.3\}$ and $\gamma \in \{0.005, 0.015, 0.15\}$. A fraction of **0.0003** of the initial population was infected initially. The probability of recovery $\epsilon$ is set to **0.005**.

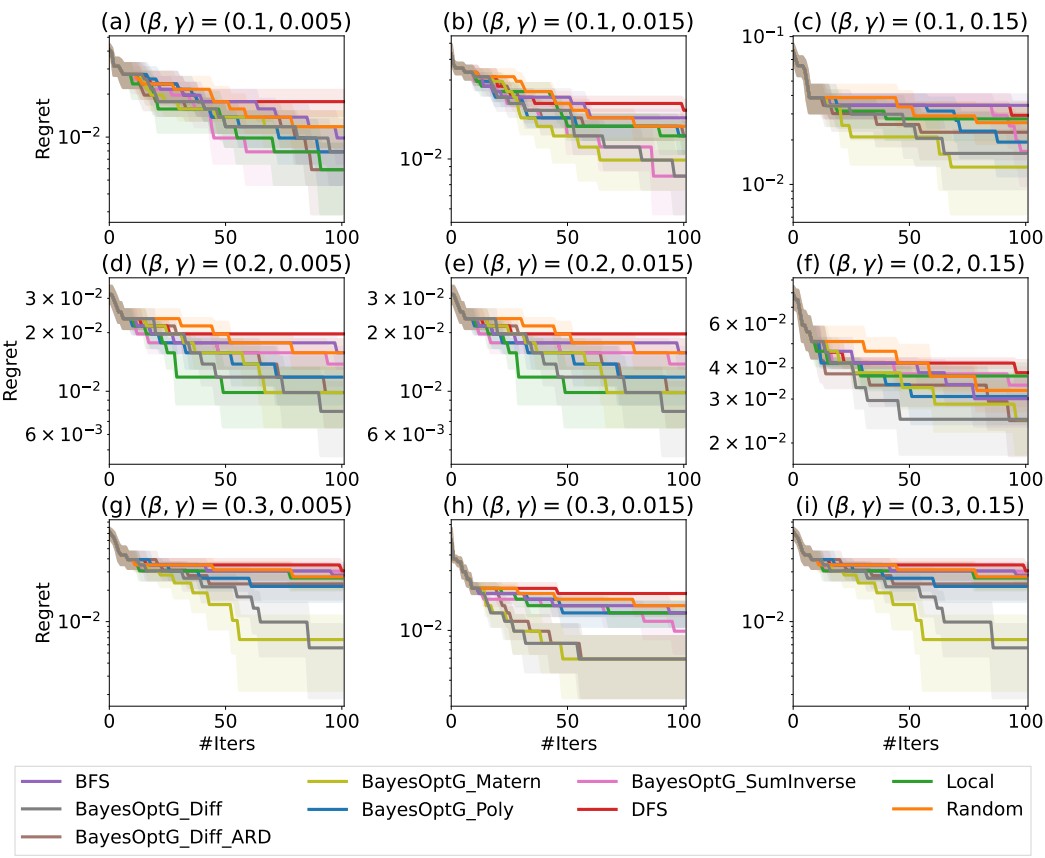

Figure 22: *Identifying the patient zero* task with different SIR model hyperparameters $\beta \in \{0.1, 0.2, 0.3\}$ and $\gamma \in \{0.005, 0.015, 0.15\}$. A fraction of **0.003** of the initial population was infected initially. The probability of recovery $\epsilon$ is set to **0**.

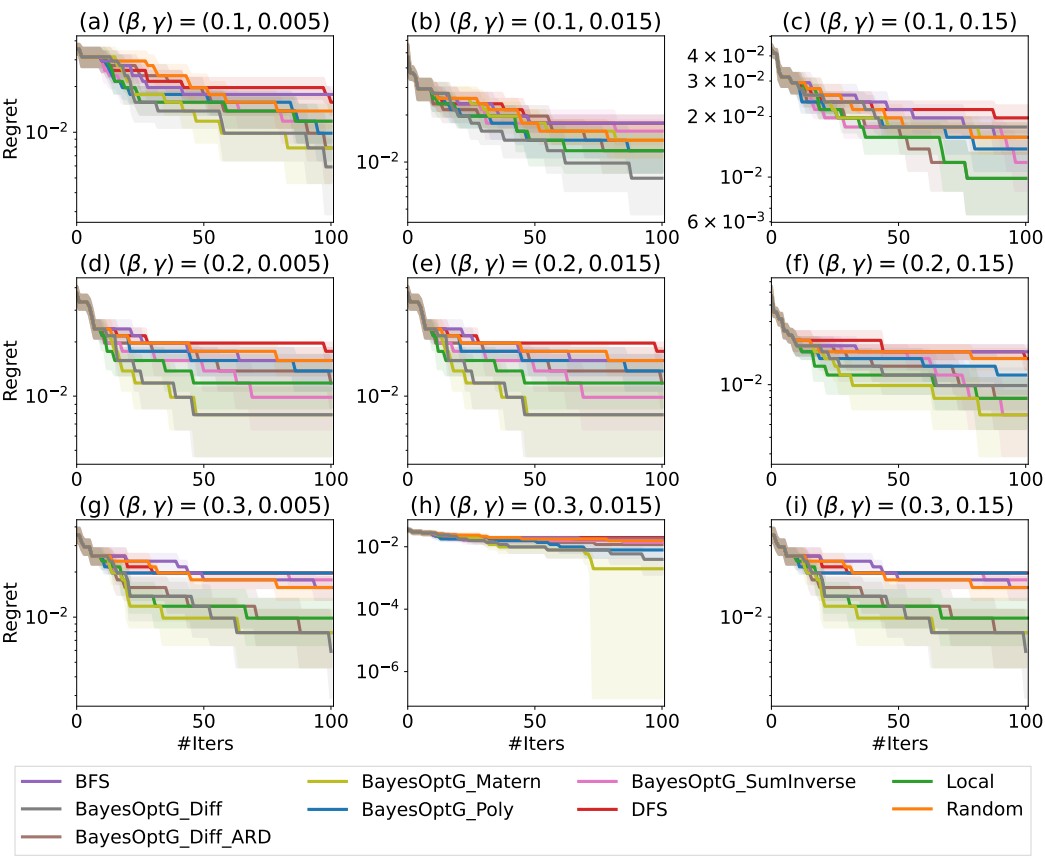

Figure 23: *Identifying the patient zero* task with different SIR model hyperparameters $\beta \in \{0.1, 0.2, 0.3\}$ and $\gamma \in \{0.005, 0.015, 0.15\}$. A fraction of **0.003** of the initial population was infected initially. The probability of recovery $\epsilon$ is set to **0.005**.

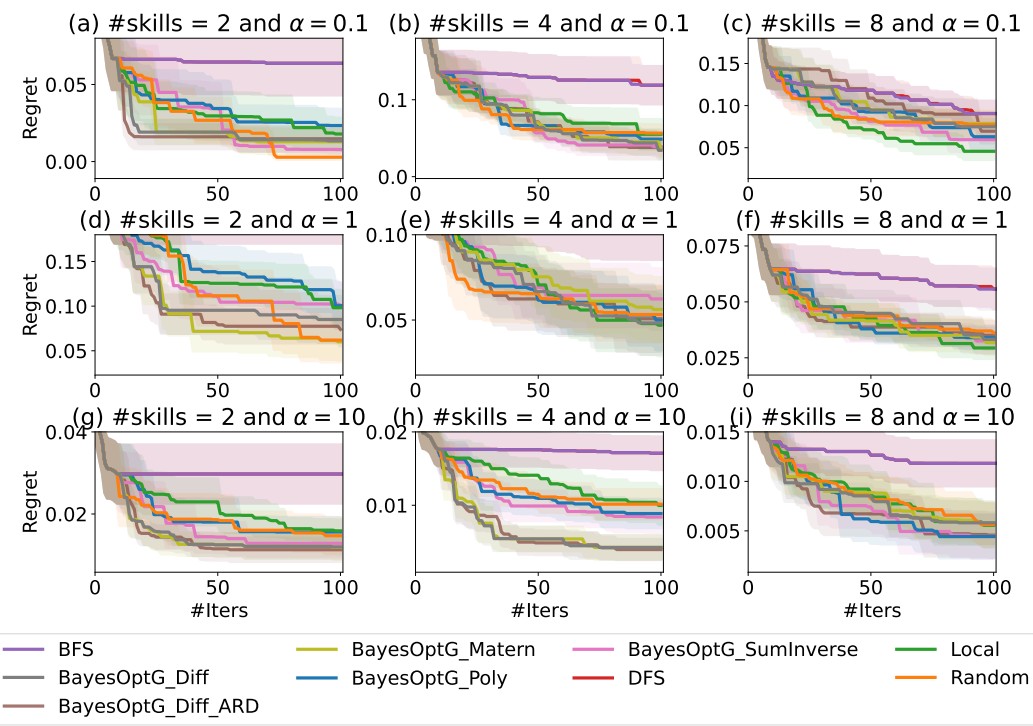

Figure 24: *Team optimisation* task with $s \in \{2, 4\}$ and $\alpha \in \{1, 10\}$ with Jaccard index threshold of **0.1** (refer to App. B.4.3 for explanations)

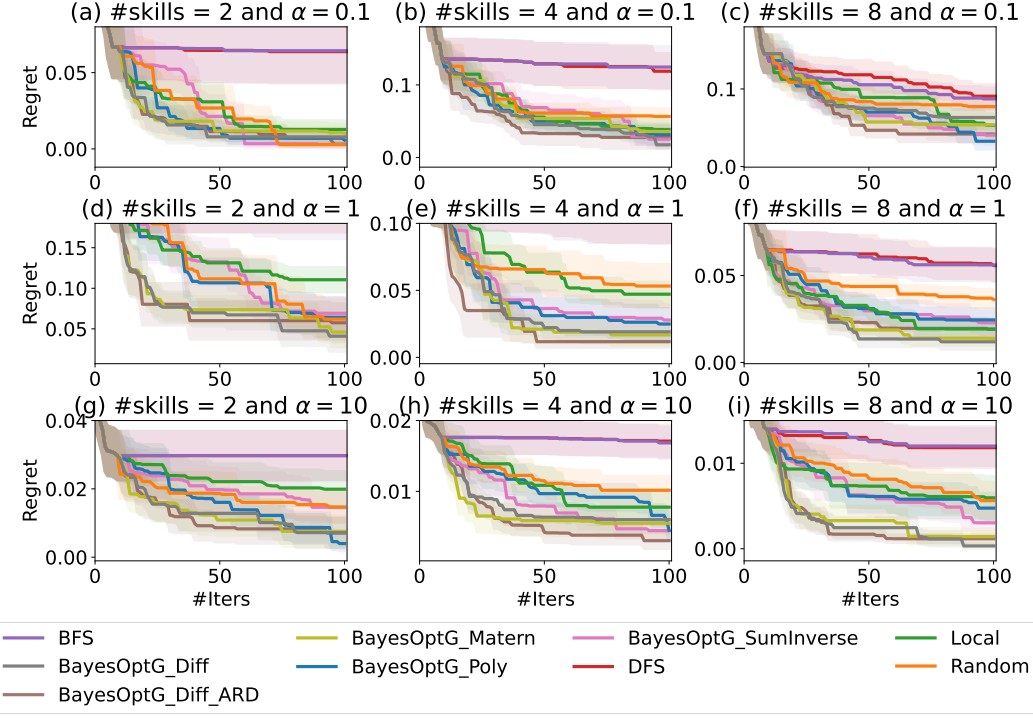

Figure 25: *Team optimisation* task with $s \in \{2, 4\}$ and $\alpha \in \{1, 10\}$ with Jaccard index threshold of **0.2** (refer to App. B.4.3 for explanations)

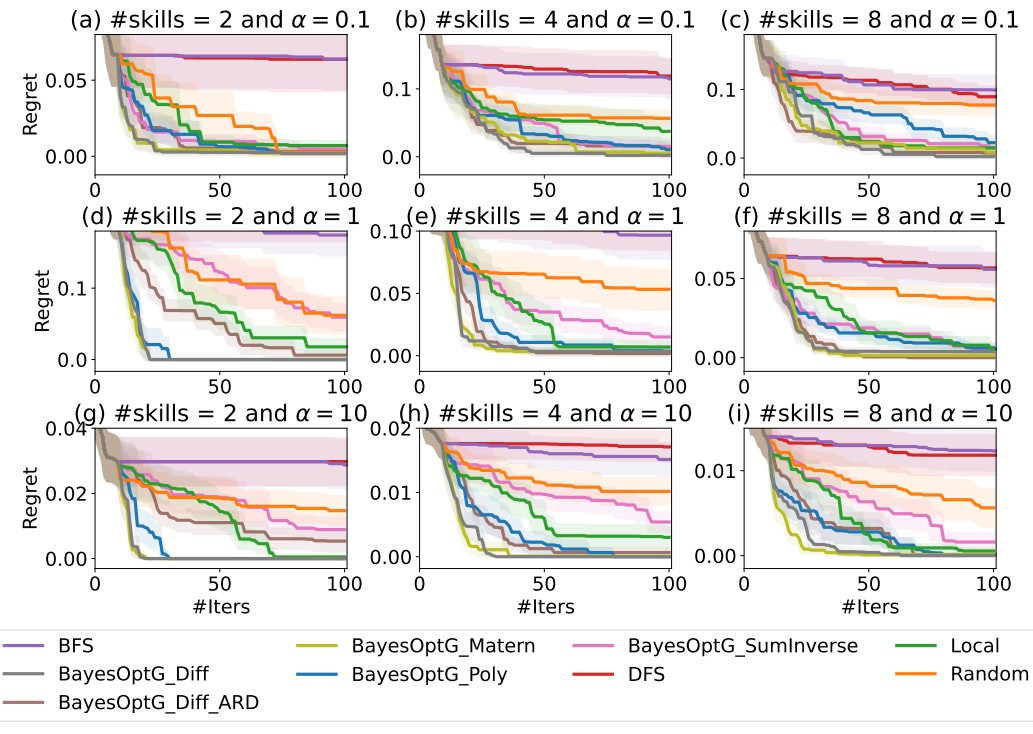

Figure 26: *Team optimisation* task with $s \in \{2, 4\}$ and $\alpha \in \{1, 10\}$ with Jaccard index threshold of **0.3** (refer to App. B.4.3 for explanations)

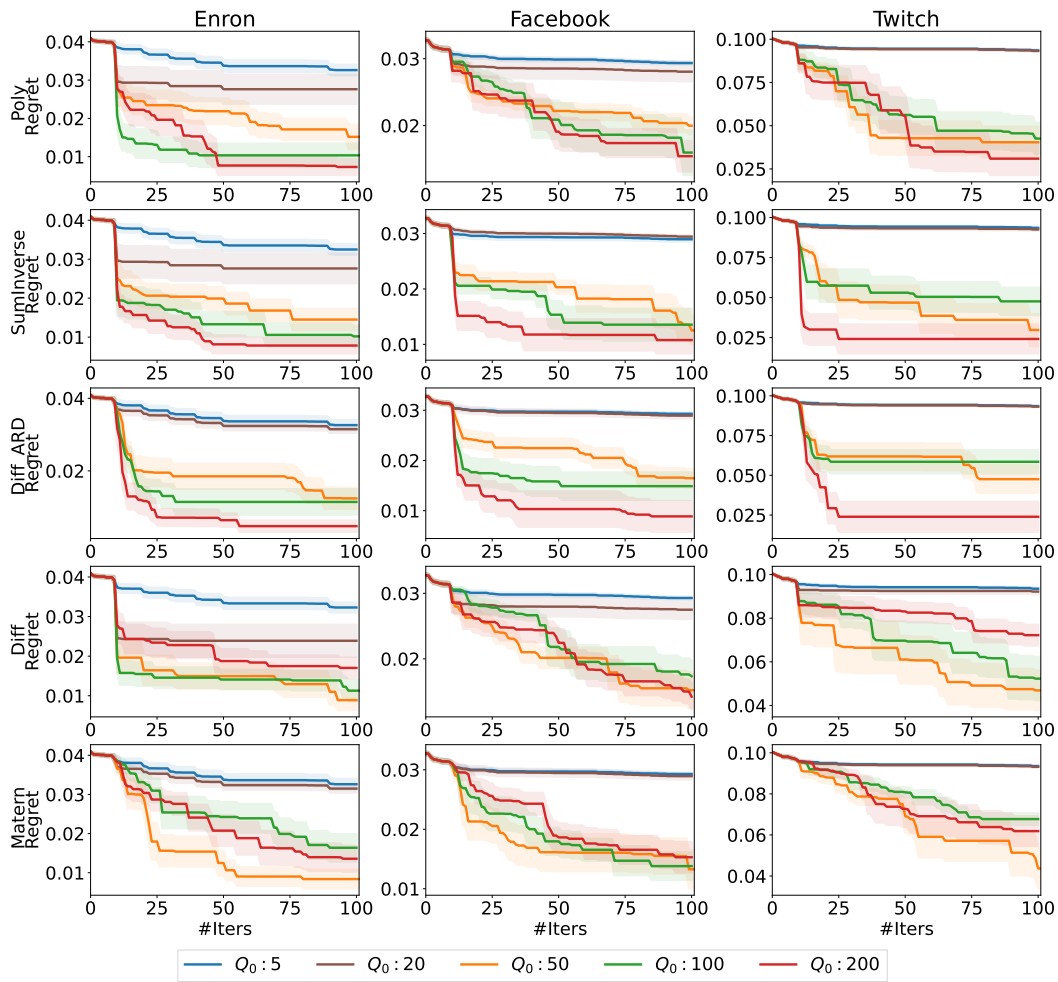

Figure 27: Sensitivity of performance to $Q_0$ on different tasks and kernels. From left to right: Centrality maximisation on Enron, Facebook and Twitch networks. Kernels from top to bottom: polynomial, sum-of-inverse polynomials, diffusion (with ARD), diffusion (without ARD), and graph Matérn.

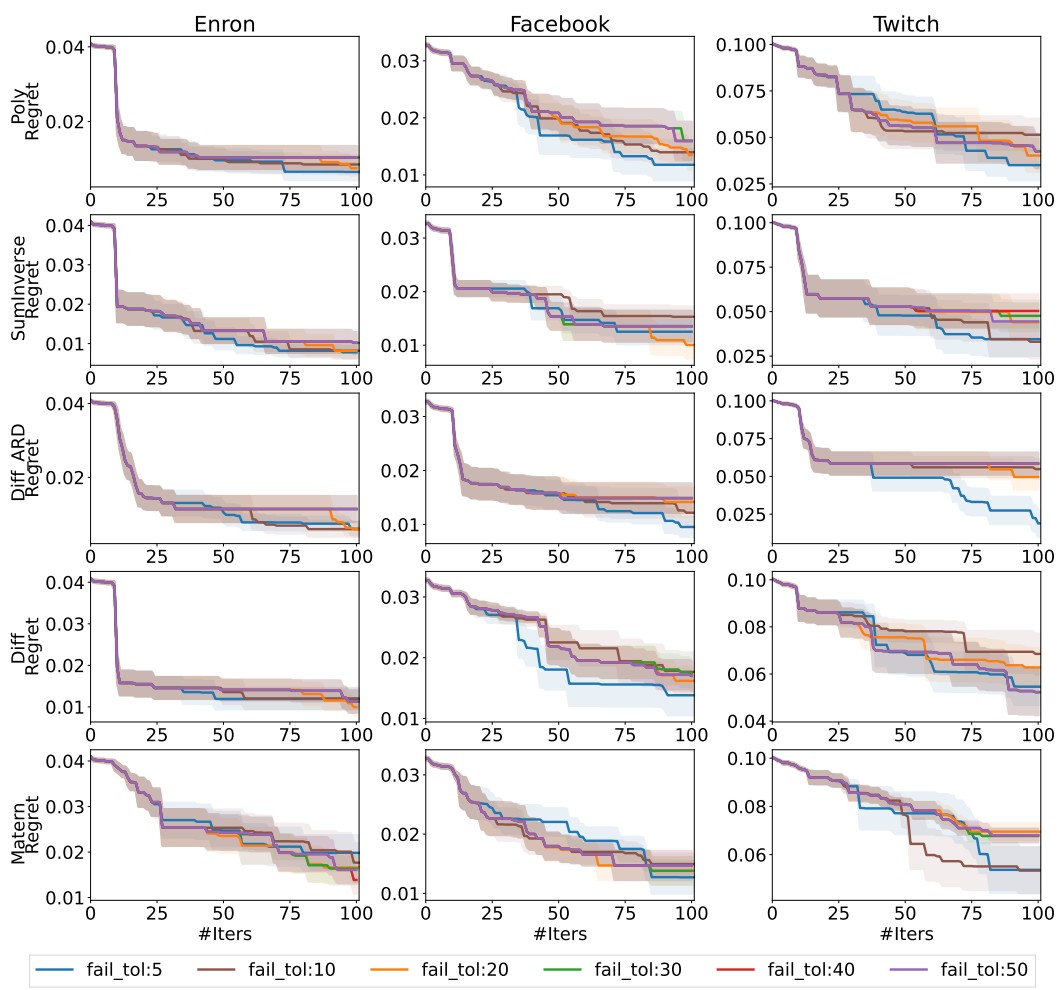

Figure 28: Sensitivity of performance to `fail_tol` on different tasks and kernels. Refer to Fig. 27 for additional explanations.

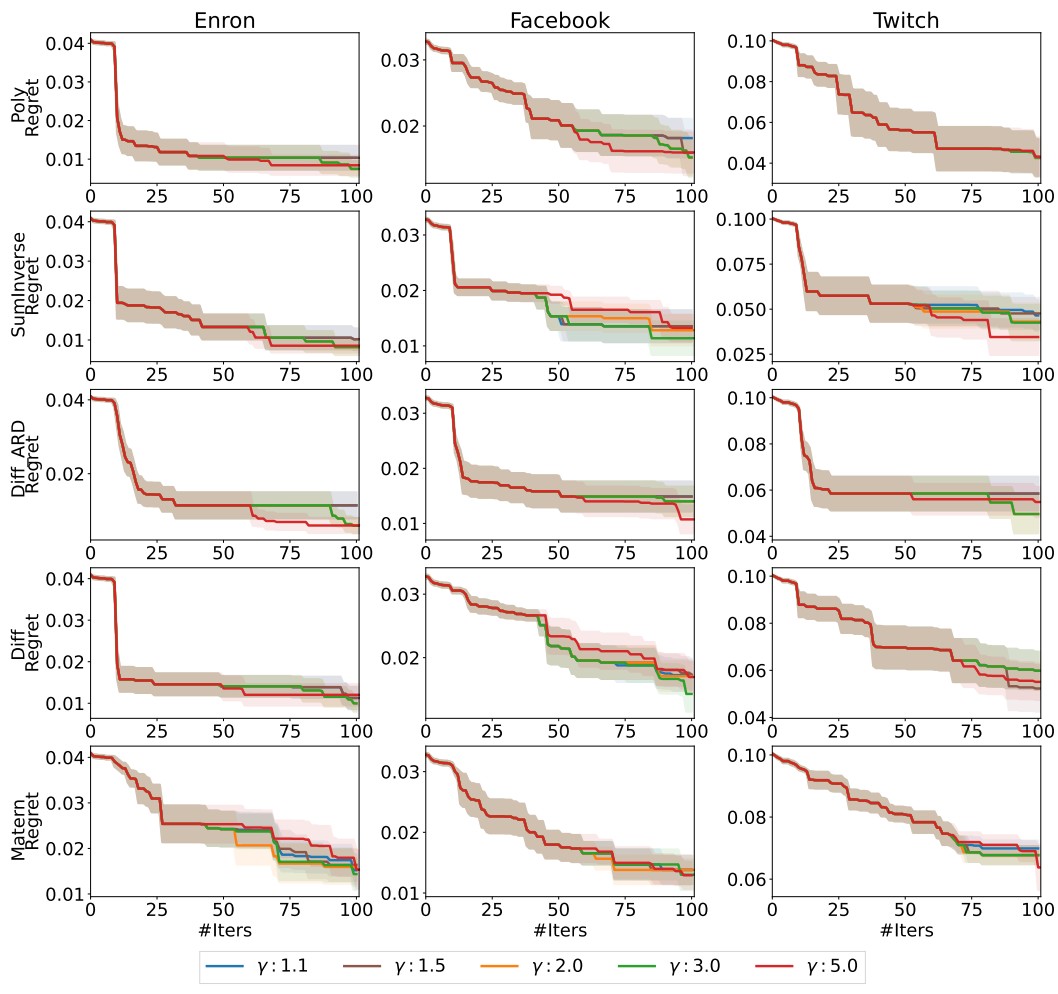

Figure 29: Sensitivity of performance to $\gamma$ on different tasks and kernels. Refer to Fig. 27 for additional explanations.

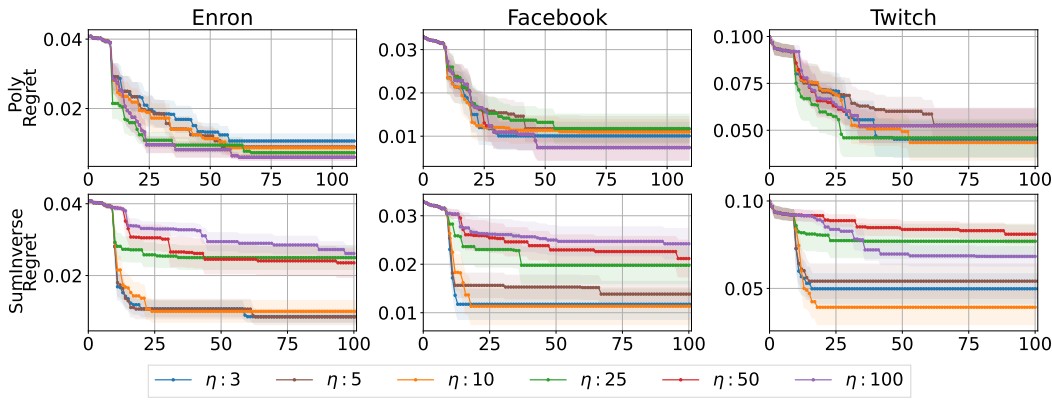

Figure 30: Sensitivity of performance to $\eta$ on different tasks and kernels. Refer to Fig. 27 for additional explanations. Note that only Polynomial and Sum-of-inverse-polynomial kernels requiring non-trivial $\eta$ selection are included.

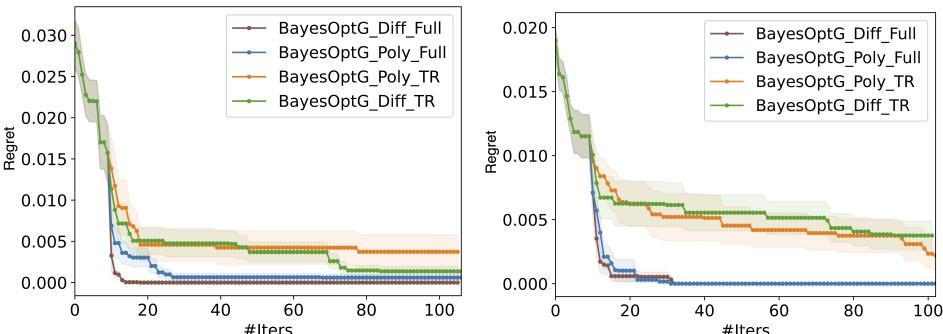

Figure 31: Influence of trust region method performance compared to full graph knowledge in a WS graph of 400 (Left) and 1000 (right) nodes. It is worth noting that while the use of trust regions leads to some trade-off in terms of optimisation performance, without trust regions, the algorithms are significantly more computationally prohibitive. In the graph of 1000 nodes (right), the computational cost is more than $20\times$ the cost of a local algorithm (shown by Fig. 3 in the main text which focuses on wall-clock time comparison).

