# OpenReview forum: "Bayesian Optimisation of Functions on Graphs"
_NeurIPS.cc/2023/Conference — NeurIPS 2023 poster_

### Official Review · Reviewer_4pXS · 2023-07-03

**Soundness:** 3 good
**Presentation:** 2 fair
**Contribution:** 2 fair
**Rating:** 5
**Confidence:** 4

**Summary:**

This article proposes a Bayesian optimization approach to solve node-level tasks with graph-structured data. The presented framework, combined with three kernels that capture the covariance functions on graph-structured data, models each node by its ego-graph of a learnable size. The authors argue that this approach not only makes the optimization process tractable but also addresses the challenge that the graph information might not be entirely available. Experiments with synthetic and real-world tasks demonstrate that the presented framework achieves superior or competitive performance.

**Strengths:**

1. Novelty: good observation on the lack of Bayesian optimization framework on node-level tasks.
2. Soundness: the three kernels, especially the latter two, are well presented. Their functions in the BO process is well explained and persuasive.
3. Clarity: Figure 2 and Algorithm 1 help the paper with information presentation. Most entities are well-defined.



**Weaknesses:**

1. Significance: BO is usually utilized to approximate black-box functions that are expensive to evaluate. However, the description in the introduction section and the tasks in the experiments fail to demonstrate the necessity of using BO.
2. Arguing the tractability of the framework, this paper did not discuss the time complexity. And there are no experiment results on running time either. (Solved)
3. Typo in Algorithm 1. 2: Output: the node that v^*_T that minimises the objective function. Extra "that", and "minimizes". (Solved)
4. Figure 2 wrong reference. "to the central node (Eq. (??) in the figure)". (Solved)

**Questions:**

If the graph information is not entirely available, how does breaking the entire graph into multiple ego graphs help? When the ego graph of a node selected in an early iteration is wrong due to missing links, the following iterations will make wrong judgments too.

---

> ### Author Rebuttal · Authors · 2023-08-09
>
>
> We appreciate the reviewer’s comments and are glad they have positively remarked on our paper’s novelty, soundness and clarity. We believe that we have addressed the reviewer’s concerns below, and in light of this, we hope that the reviewer will consider increasing their rating.
>
> > Significance: BO is usually utilized to approximate black-box functions that are expensive to evaluate. However, the description in the introduction section and the tasks in the experiments fail to demonstrate the necessity of using BO.
>
> We will improve the presentation in Introduction and Experiments as suggested and refer the reviewer to our [overall response](https://openreview.net/forum?id=UuNd9A6noD&noteId=FsjM9qNZ9q) on this point.
>
> > Arguing the tractability of the framework, this paper did not discuss the time complexity. And there are no experiment results on running time either.
>
> We include the time complexity analysis below, which we will add to the final version of the paper.
>
> Firstly, as with all GP-based BO algorithms, BayesOptG scales cubically with the number of training points $N$: $\mathcal{O}(N^3)$, assuming no other efficient approximations are used. A unique challenge in the graph case is that the algorithm can also be bottlenecked by the *search space size*, as computing the kernels requires eigendecomposition of the Laplacian matrix of the graph, a basic algorithm for which scales $\mathcal{O}(n^3)$ where $n$ is the number of nodes. Thus, the overall complexity can be given by $\mathcal{O}(N^3 + n^3)$.
>
> The restart and local modelling algorithm introduced in §3.2 ensures both terms are tractable. By placing the GP over the local subgraph only, $n$ is now upper limited by the max subgraph size instead of the size of the entire graph, which can be very large. By periodically restarting the GP when the optimisation gets stuck, we implicitly prevent $N$ from growing very large. This is why local modelling ensures the tractability of the framework.
>
> We also add the running time comparison the reviewer requested in Fig S6 of the rebuttal PDF to demonstrate this more concretely. It is, however, worth noting that exactly due to what the reviewer mentioned previously, given that BO is commonly used to optimise expensive functions, in real life, the computational cost is likely dominated by the cost of querying the objective functions, and thus the number of objective function evaluation that we used in the paper is often a better proxy of the overall cost.
>
>
> > Typo in Algorithm 1. 2: Output: the node that v^*_T that minimises the objective function. Extra "that", and "minimizes".
> > Figure 2 wrong reference. "to the central node (Eq. (??) in the figure)".
>
> We thank the reviewer and will correct these typos. We were trying to refer to the mathematical definition of the central node (Line 209).
>
> > If the graph information is not entirely available, how does breaking the entire graph into multiple ego graphs help? When the ego graph of a node selected in an early iteration is wrong due to missing links, the following iterations will make wrong judgments too.
>
> We believe this is a misunderstanding in our setup, and we will clarify this crucial aspect when we revise the paper.
>
> When we mentioned that “graph information is not entirely available”, we refer to the setup that we cannot access the graph in full – this is very common either due to the size (e.g., we cannot access the entire Facebook graph as it is too large) or cost (e.g. in the contact tracing example, it is impractical to have the full information beforehand, we that would require us to interview and contact-trace everyone involved). What we can do is query a node and reveal the graph structure around it (the ego-network part), *noiselessly* – this is also a reasonable assumption: using the examples above, this is akin to viewing *a specific person’s* Facebook friends and conducting an exhaustive contact-tracing interview on *a specific person*. Crucially, at this step, we do not consider the case that the network structure revealed locally is noisy or otherwise erroneous (e.g. with missing or fake links), nor are we “breaking the entire graph into multiple ego graphs” as the reviewer suggests – we are simply revealing the structure on the fly.
>
> In other words, when we say “graph information is not entirely available”, we mean that the global graph is not available to the BO agent initially, but it may query nodes and reveal local subgraphs around them when the optimisation proceeds. This does not mean we have a partial, noisy or otherwise inaccurate graph structure to begin with, which seems to be what the reviewer suggests. While we agree that studying such a more challenging setup is interesting and will include it in future works, this is out-of-scope of the current submission (especially in view that the current submission is one of the first attempts of using BO in this setup, as the reviewer also agreed on).

---

> > ### Comment · Reviewer_4pXS · 2023-08-17
> >
> > Thanks for your effort. I appreciate your presentation about the time complexity. For the necessity of using BO, I hope this can be identified more clearly in the later version. Rating increased to 5.

---

> > > ### Author Response · Authors · 2023-08-21
> > >
> > > We thank the reviewer for their feedback and will make sure to incorporate suggested changes in the final version of the manuscript.

---

### Official Review · Reviewer_GnE3 · 2023-07-06

**Soundness:** 3 good
**Presentation:** 2 fair
**Contribution:** 2 fair
**Rating:** 4
**Confidence:** 3

**Summary:**

The authors consider the Bayesian optimization for functions defined on a graph (e.g., finding a node in a graph to min/max some function on that graph). The authors propose a local modeling approach for such problem on generic, large-scale and potentially unknown graphs. The authors demonstrate the advantages of the proposed approach on several experiments.

**Strengths:**

+ It is interesting to consider Bayesian optimization approach for a function on a graph (i.e., finding a node in a graph which minimizes/maximizes some function defined on that graph).
+ The local modeling makes the approach to scale up for large-scale graph, potentially unknown graph.
+ The authors also propose two kernels for Bayesian optimization for functions on graph (which are variant of the diffusion kernel on graph)
+ The authors provide extensive experiments for the proposed approach.

**Weaknesses:**

+ The experiments seem weak. It is unclear whether the authors have considered any large-scale graphs, or potentially unknown graphs in the experiments from the main manuscript. It is also unclear about the objective functions for all the experiment whether one needs Bayesian optimization for the tasks yet.
+ Some important part needs to elaborate with details, e.g., especially selecting the local subgraph; why the proposed approach can handle potential unknown graph?

**Questions:**

The flow of the submission is clear. However, it is better in case the authors elaborate more details for some important parts.
+ Although the authors discuss in line 229-241, the proposed approach (e.g., local modeling) is closely related to the trust-region-BO. In my understanding, the local modeling is at the heart to make the approach handle possibly large-scale graph. In some sense, they share the same role.
+ The authors should elaborate how to select the next “subgraph” (even its size is given). Does the selected subgraph require to be connected? (since it seems that the authors do not assume that the graph is connected?). Is there any overlap between subgraphs at different iterations?
+ The authors should give more details for their experiments in the main manuscript, especially the size of the graph and the objective function for optimization. It is unclear that the authors illustrate the proposed approach for large-scale graphs, (or potentially unknown graphs?) to optimize functions on graph (which are expensive to evaluate) yet.
+ It is unclear how the proposed algorithm can handle unknow graphs as claimed? Could the authors elaborate it with more details?
+ It is better to elaborate more details about the Algorithm 1 and 2 in the main text, especially the Algorithm 2.
+ For the strategy as in line 205-215, could the authors give more explanation when the algorithm finds a subgraph with local minima, is it stuck on that region around the subgraph, or it has some strategy to improve the objective function more? (e.g., by explore to some far regions?)

Some minor points:
+ What is $\tilde{A}$ in line 140? There is no explanation for it.
+ Reference for Equation in Figure 2

---

I thank the authors for the rebuttal.

**Limitations:**

The authors have not discussed the limitations and potential negative societal impact of their work.

---

> ### Author Rebuttal · Authors · 2023-08-09
>
> We thank the reviewer for their detailed feedback. It seems that the primary concern is 1) how our method deal with unknown graphs and 2) some details of BayesOpt. We address both below, and we hope the reviewer will consider increasing their rating in light of our response.
>
> > The experiments seem weak.
>
> First, as the reviewer acknowledged, our paper features an extensive experimental section, perhaps more so than many published works in BO, even though we are considering a novel setup with scarce prior works. Second, BayesOptG consistently outperforms the baselines despite the differences in underlying function properties. We believe both strongly support the strength of our experiments.
>
> > Large-scale & unknown graphs
>
> “Size”: We considered varying graph sizes and detailed them in the figure captions ranging from thousands (synthetic tasks) to tens of thousands in Fig 9 (34,000 for Enron, 14,000 for Facebook graph). In some experiments, the use of smaller graphs (of $\mathcal{O}(10^3)$) is attributable to the cost of computing the objective function (e.g. betweenness or eigenvector centrality) and not because our method cannot be used for large graphs. To illustrate this, we scale BayesOptG to 1,000,000-node graphs for some tasks in Fig S3 of the rebuttal PDF, where it retains strong performance.
>
> “Potentially unknown graphs”: for all experiments, the full information about the graphs is *never* revealed a-priori (except for regression experiments in §5.1 to test the predictive powers), and thus from this perspective, all experiments feature “unknown” graphs to the BO agent at the start of the algorithm. The graph structure is only incrementally revealed to the BO as the optimisation proceeds: when BO queries a point, its ego-network, a local subgraph, is revealed.
>
> > Why one needs BO for the tasks
>
> Please see the overall response.
>
> > details on local subgraph selection and why this handles unknown graphs
>
> The subgraph selection is elaborated by Algorithm 2 and §3.2 – the approach may handle unknown graphs as it does not require the full graph information to be known a-priori. Only when the BO queries a node the graph structure of the ego-network centred around that node is revealed to the BO agent. As such, the algorithm works as long as we can request to reveal the local information around the nodes we query.
>
> > Although the authors discuss in line 229-241, the proposed approach (e.g., local modeling) is closely related to the trust-region-BO. In my understanding, the local modeling is at the heart to make the approach handle possibly large-scale graph. In some sense, they share the same role.
>
> We’d be grateful if the reviewer could clarify the question in this remark, but we will explain what we meant in paragraph *Remarks on the relation to trust-region BO methods* in §3.2.
>
> The reviewer is right to say that the local modelling makes scaling the method to large graphs possible, as we only impose GPs on the subgraph. However, as discussed above, local modelling also makes it possible to apply the algorithm to unknown graphs as, at any point in time, the algorithm *only requires information about this local graph* and works as long as we are capable of collecting and querying local graph information around the queried node.
>
> It’s also worth noting that the primary objective of trust regions in previous BO algorithms is *not* to ensure scalability but to alleviate the curse of dimensionality of GP by focusing on a promising subspace to reduce over-exploration [1]. In the typical BO setup, the scalability is typically purely bottlenecked by the number of training points. Unlike our graph setup, the search space size *plays no role in scalability*. While trust region BO, by the periodic restarting of the GP, resets the training samples of the GP & is indeed more scalable than an algorithm that does not restart, we argue this is a side-effect of restarting and not of trust regions. We argue this is another difference in motivation between us and previous trust region BO methods.
>
> [1] Eriksson et al. (2019). Scalable global optimization via local Bayesian optimization. NeurIPS.
>
> > how to select the next “subgraph”
>
> The procedure to select the next subgraph is in Algorithm 2. As mentioned in Line 209, the subgraph is constructed around the node with the best objective function seen so far.
>
> > Does the selected subgraph require to be connected?
>
> The subgraph is the ego-network centred around the best node seen so far, so it is, by definition, connected. Our algorithm still works when there are disconnected components due to the restart mechanism (line 227): while within one restart, the BO will explore one connected graph, the BO may jump to another disconnected subgraph after a restart since the restart location is determined randomly.
>
> > Overlap between subgraphs at different iterations
>
> Yes – if the best objective value did not increase, the subgraph does not change from the previous iteration. Otherwise, it is moved to be centred around the new best node, but there will still be overlap (as the new best node must lie within the preceding subgraph) – as illustrated in Fig 1. An additional point to clarify is that the GP is always imposed over the *current* subgraph: when the subgraph changes, some evaluated nodes will no longer be in the current subgraph. We do not impose GP over these points, so the subgraph does not accumulate over previous evaluations.
>
> > is the algorithm stuck on that region around the subgraph, or does it have some strategy to improve  more?
>
> Please see our description of the restart mechanism to deal with this issue in Lines 227-228.
>
> > tilde{A}
>
> It is the adjacency matrix of the subgraph. We will clarify this in the revised manuscript.
>
> > Equation in Figure 2
>
> We will address this in the revised manuscript. We meant to refer to the mathematical definition of the central node, i.e. the best node seen so far (Line 209).

---

> > ### Author Response · Authors · 2023-08-10
> > **Clarification on the definition of "scalability" in the rebuttal**
> >
> > We'd like to provide a quick clarification regarding *scalability* when we discuss previous works in BO:
> >
> > When we argued that "the main objective of trust regions in prior BO works is not for scalability" in the rebuttal, we were referring to the scalability in terms of *the number of training points* -- this is also the definition used in seminal papers like [1]. However, if we use a broader definition to include scalability to *the number of dimensions*, we agree that trust regions also help scalability in prior BO literature as they improve the performance of GPs in high dimensions.
> >
> > Despite any definitional differences, our core argument still stands: in prior BO methods, trust regions are mainly used to improve GP performance in high dimensions rather than to accommodate more data. In our case, local subgraphs achieve both: they both restrict the GP's attention to a promising subregion similar to trust regions but also make BO more scalable to more training points, as the GP defined on graphs also scale cubically w.r.t the number of nodes $n$, in addition to the number of training points $N$ with $\mathcal{O}(n^3+N^3)$. Using subgraphs ensures $n$ is tractable -- empirical evidence of this is provided in Fig. S6 of the rebuttal PDF.
> >
> > We thank the reviewer once again for their comments.
> >
> > ### References
> > [1] Liu, H., Ong, Y. S., Shen, X., & Cai, J. (2020). When Gaussian process meets big data: A review of scalable GPs. IEEE transactions on neural networks and learning systems, 31(11), 4405-4423.

---

> > ### Comment · Reviewer_GnE3 · 2023-08-18
> >
> > Thank you for the rebuttal. I have no other raised points.

---

> > > ### Author Response · Authors · 2023-08-19
> > >
> > > We thank the reviewer again for engaging in the rebuttal process. We hope that our response helped clarify the points raised by the reviewer and ease the understanding of our paper, and we are glad that the reviewer has no further outstanding concerns.
> > >
> > > If there is improvement that we can still make at this stage that would make the reviewer evaluate our work more positively and raise their score, we would be very happy to do so. Thank you!

---

### Official Review · Reviewer_tMg5 · 2023-07-06

**Soundness:** 2 fair
**Presentation:** 2 fair
**Contribution:** 2 fair
**Rating:** 3
**Confidence:** 4

**Summary:**

This paper solves an optimization problem defined on a graph.  Since the problem defined on a graph requires the need to search for a solution in a combinatorial manner, it is a challenging problem.  To tackle such a problem, the authors investigate diverse kernels on a graph and a local search strategy on a graph.  Finally, they provide the experimental results that show the performance of the investigated methods.

**Strengths:**

* The problem defined on a graph has not been studied much.  This topic in particular is interesting in the Bayesian optimization community.
* Experiment section contains many things to discuss.

**Weaknesses:**

* Analysis on which method is strong at some circumstances and which factor affects to the performance is lacked.
* Analysis on the investigated methods is lacked.
* Discussion on local search strategies is weak.
* Experimental results seem not consistent.
* Writing can be improved.

**Questions:**

1. As described above, the authors can provide more thorough discussion on which method is strong at some circumstances and which factor affects to the performance.
2. Moreover, the authors can discuss analysis on the investigated method in terms of kernel types and local search strategies.  For example, a diffusion kernel is generally able to capture meaningful relationship between two nodes compared to polynomial and sum of inverse polynomials kernels.  On the contrary, the other kernels show their effectiveness in some conditions.  The discussion like this should be included in this paper in order to understand the algorithms appropriately.
3. Basically, the experimental results are hard to interpret.  Figures are too small and lines are too thin.
4. The number of evaluations is too small.
5. Regardless of the visualization of the experimental results, there is no consistency I think.  I do not much care about the inconsistency, but the authors need to discuss why it happens.
6. I cannot understand the experiment of team optimization.  Each node represents a team, right?  How many team members belong to a team?  The size of team does not affect to the evaluation of team?  I think a skill vector of each member is sum to one because it is drawn from the Dirichlet distribution.  But if the sum of skill sets for team members exceeds 1, what happens?  Is it just treated as 1 or the sum of skill values?

**Limitations:**

I do not think that this work has specific limitations.

---

> ### Author Rebuttal · Authors · 2023-08-09
>
> We thank the reviewer for their insightful comments. It seems to us that the biggest concern comes from the discussions of the experimental results and the relative strengths of different methods in different situations. We agree with the reviewer that such a discussion would be helpful for users, especially in a practical setup, and we thank the reviewer for their suggestions. We include a detailed discussion on this point in the [overall response](https://openreview.net/forum?id=UuNd9A6noD&noteId=FsjM9qNZ9q), which will be incorporated into the manuscript, and we are confident that the issues pointed out in the review can be fixed when we update the manuscript. We hope the reviewer can consider increasing their rating in light of this.
>
> > Experimental results seem not consistent. The authors need to discuss why it happens.
>
> While we refer the reviewer to our overall response, we would like to emphasise that *our proposed algorithm, BayesOptG* (marked as “BO\_*” in the figures), *always outperforms or at least performs on par compared to the best baseline* (except for Fig 4 if we opt to use obviously inappropriate kernels such as BO_Diff (without ARD) against the local search, the strongest baseline – see our explanations below). **From this perspective, there is little inconsistency in experimental results**. While there are indeed differences between different kernel choices, as we discussed in the overall response, the fact that different kernels are suited for different problems and that the kernel choice often significantly impacts the modelling performance of GPs is highly expected for *any* GP-based technique. We argue that this is not a weakness of our method, but the fact that BayesOptG performs strongly *regardless of the kernel choice* demonstrates its robustness.
>
> > Analysis on the investigated methods is lacked.
>
> > As described above, the authors can provide more thorough discussion on which method is strong at some circumstances and which factor affects to the performance.
>
> > Analysis on which method is strong at some circumstances and which factor affects to the performance is lacked.
>
> > Discussion on local search strategies is weak.
>
> > Moreover, the authors can discuss analysis on the investigated method in terms of kernel types and local search strategies. For example, a diffusion kernel is generally able to capture meaningful relationship between two nodes compared to polynomial and sum of inverse polynomials kernels. On the contrary, the other kernels show their effectiveness in some conditions. The discussion like this should be included in this paper in order to understand the algorithms appropriately.
>
> As we mentioned at the beginning of the response, we thank the reviewer for the suggestion and agree this can be important for potential users. We direct the reviewer to our “Overall Response” for the requested discussions in response to all the comments above that the reviewer raised.
>
> > Basically, the experimental results are hard to interpret. Figures are too small and lines are too thin.
>
> We thank the reviewer for this feedback. We will address this in the revised manuscript.
>
> > The number of evaluations is too small.
>
> We are unsure whether the reviewer meant i) the query budget or ii) the number of random trials.
>
> If the reviewer meant i): while we agree it is advantageous to run longer experiments, it is not uncommon in BO work to set the query budget around 100, given the assumption that the evaluation is expensive. If the reviewer meant ii), we again agree that increasing the number of random trials would always be beneficial. However, we argue that the BO algorithm's outperformance over the baselines is strong and consistent with the 10 repetitions we used already.
>
> In any case, we thank the reviewer’s suggestion and provide some additional experiments where we increase both the query budget and the number of random trials, and the reviewer is referred to Fig S3 and S4 in the rebuttal pdf where the paper's main findings still stand.
>
> > I cannot understand the experiment of team optimization. Each node represents a team, right? How many team members belong to a team? The size of team does not affect to the evaluation of team? I think a skill vector of each member is sum to one because it is drawn from the Dirichlet distribution. But if the sum of skill sets for team members exceeds 1, what happens? Is it just treated as 1 or the sum of skill values?
>
> We thank the reviewer for the opportunity to clarify our work. We initially set a maximum number of members in team $N$, and a number of “skills” $K$. Then, the graph is built such that each node represents a team with less than $N$ members (there are $2^N$ of these, hence potentially very large), and an edge is defined between teams that have a minimum number of common members (above a certain quantile of the distribution of common members across all possible pairs of teams). The objective function aims at maximising team competence by promoting members who are experts, and that complement each other. One member is indeed a discrete distribution over skills, and so is each team, where each skill component is the average skill component over its members, hence also defining a distribution. If we define the objective function by the difference between the entropy of the skill distribution of the team and the average entropy of the skill distribution over its members, the objective promotes the completeness of the team and the sparsity of skills among the team members. Regarding the size of the team, the reviewer is right in stating it does not influence the objective. We will address this issue by adding a regularising term in a revised version of the manuscript.

---

> > ### Comment · Reviewer_tMg5 · 2023-08-17
> >
> > Thank you for your response.  I acknowledge that I have read your rebuttal.
> >
> > After reading the rebuttal, I decided to keep my rating.

---

> > > ### Author Response · Authors · 2023-08-17
> > > **Could you please elaborate?**
> > >
> > > We appreciate the reviewer’s acknowledgment, but we’d be grateful if the reviewer could explain why they insist on a rejection recommendation.
> > >
> > > As mentioned, we believe the reviewer’s primary concerns were 1) analysis of which method is strong in some circumstances and factors affecting performance and 2) consistency of the results. We firmly believe that we addressed both points. Other concerns include a number of evaluations and clarification on team optimization, and again we provided thorough responses, sometimes through additional experiments.
> > >
> > > If the reviewer feels that we have not addressed their concerns satisfactorily, we’d be grateful if they could comment on what, in their opinion, is lacking, and we will try our best to respond.
> > >
> > > Again, we appreciate the reviewer’s help in improving the quality of our work.

---

### Official Review · Reviewer_sVur · 2023-07-07

**Soundness:** 4 excellent
**Presentation:** 3 good
**Contribution:** 3 good
**Rating:** 8
**Confidence:** 3

**Summary:**

The paper presents a Bayesian optimization algorithm for functions defined on the nodes of (potentially unknown) graphs.  The algorithm combines local modelling via trust regions to account for the potentially unknown nature of graphs with random restart to avoid becoming stuck in local minima.  Novel kernels on graphs are defined (locally) to avoid problems related to overfitting.

**Strengths:**

This paper addresses a novel problem, the discussion is compelling and the results appear to be sound.

**Weaknesses:**

(see questions below).

Minor point: figure 2 - equation (??).

**Questions:**

- Step 4 in algorithm 1: it is unclear how this step is to be interpreted on the first iteration of the algorithm with $v_t^*$ is not defined yet.
- Why set $Q_{\rm min} = 1$ (line 226)?  Wouldn't this mean that you *only* test $v_t^*$ for that iteration (region size of $1$), which is redundant as you already (presumably) know the value of $f$ at this node?

---

> ### Author Rebuttal · Authors · 2023-08-09
>
>
> We thank the reviewer for their positive and insightful impact and are glad the reviewer found our discussions novel, compelling and sound. Please see below for our response to the concerns the reviewer raised.
>
> > Minor point: figure 2 - equation (??).
>
> We thank the reviewer for pointing this out. We were referring to the mathematical definition of the central node (Line 209).
>
> > Step 4 in algorithm 1: it is unclear how this step is to be interpreted on the first iteration of the algorithm with $v^*_t$ is not defined yet.
>
> The reviewer is correct in pointing out this fact. This should instead be placed after Line 8 (i.e. after “end if”). $v^*_t$ is initially found by finding the best point in all the randomly initialising points.
>
> > Why set  Q_min = 1(line 226)? Wouldn't this mean that you only test v_t^* for that iteration which is redundant as you already (presumably) know the value of at this node?
>
> The reviewer is correct in pointing out that when $Q=1$, we have a trivial subgraph consisting of only the node we have queried. We set $Q_{min}$ to 1, as when this happens, we know, *with certainty*, that the current subgraph contains no more unknown information and that a better solution definitely does not exist. In practice, we restart when $Q_{min}$ = 1 or all nodes in the subgraph have been queried before. It is indeed possible to set $Q_{min}$ to a value larger than 1, but that creates an additional hyperparameter; we set it to 1 because, as discussed, this is one of the most definite signals requiring a restart.
>
> We will clarify this when we update the manuscript.

---

> > ### Comment · Reviewer_sVur · 2023-08-18
> >
> > I would like to thank the reviewer for their response.  I have no further questions and will be keeping my evaluation unchanged.

---

> > > ### Author Response · Authors · 2023-08-21
> > >
> > > We thank the reviewer for engaging in the rebuttal and for their time and effort in helping to improve our manuscript.

---

### Official Review · Reviewer_nGA9 · 2023-07-26

**Soundness:** 3 good
**Presentation:** 3 good
**Contribution:** 3 good
**Rating:** 7
**Confidence:** 4

**Summary:**

The paper extends the use of Bayesian optimization based methods for optimization of the functions over the nodes of graph this algorithm is dubbed as BayesOptG. Paper mainly focuses on following three aspects:
1. Kernel design - Authors introduce suitable kernels, diffusion kernel, polynomial kernel and sum of inverse polynomial kernel, which can be used with the Gaussian process (GP) surrogate.
2. BayesOptG - Algorithm uses GP surrogate with suitable kernels to model the function on graph followed by the use of standard acquisition functions to determine the next query, the algorithm adapts the idea of trust regions not only to scale to larger graphs but also to handle imperfect knowledge about the graphs.
3. Empirical Results - The algorithm is empirically evaluated both on synthetic and real world datasets and is compared to the other baselines such as random search, local search, depth first search, breadth first search.

**Strengths:**

1. This work extends the BO based methods to the functions defined over the nodes of graph.
2. The algorithm introduced is scalable to larger and general graphs, further it can also be used in the cases with imperfect knowledge of the graph.
3. Clear introduction of the problem setup and where exactly the BO is being applied.
4. Experimental evaluation of the algorithm both on synthetic and Real world datasets.

**Weaknesses:**

Though the paper provides good experimental evaluations for the algorithm it lacks on the theoretical results. Further, little to no intuition is provided why the given choices of kernel functions are the right ones and the guarantees on the semi positive definiteness of the covariance matrix is also missing. The experiments section can further be improved by comparing the results with other algorithms such as spectral bandits[1], and GRUB [2].


[1] Valko, Michal, et al. "Spectral bandits for smooth graph functions." International Conference on Machine Learning. PMLR, 2014.

[2] Thaker, Parth, et al. "Maximizing and Satisficing in Multi-armed Bandits with Graph Information." Advances in Neural Information Processing Systems 35 (2022): 2019-2032.

**Questions:**

Questions -
1. Can you please provide a bit of background on why the kernels mentioned in the paper would result in semi positive definite covariance matrices?
2. Why are the kernels chosen the right choice and what is the intuition behind the choice? Can the Matern kernel introduced in [3] be used?

Suggestions -
1. Check the captions of Figure 2 and caption of figure 3 is hard to follow.
2. Few acronyms were introduced after they were used in prior sections. ex. BA and WS in section 5.1.
3. Section A2 introduction BFS and DFS is probably jumbled.

[3] Borovitskiy, Viacheslav, et al. "Matérn Gaussian processes on graphs." International Conference on Artificial Intelligence and Statistics. PMLR, 2021.

---

> ### Author Rebuttal · Authors · 2023-08-09
>
> We thank the reviewer for their insightful and positive feedback. We are glad the reviewer commented positively on our method's strength and experimental results. Please see below for our response to the reviewer’s concerns.
>
> > little to no intuition is provided why the given choices of kernel functions are the right ones
> > Why are the kernels chosen the right choice and what is the intuition behind the choice?
>
> We thank the reviewer for the suggestions on adding these discussions and insights, which we agree would be useful for prospective users of our algorithm. Please see the [Overall Response: Discussions of relative strengths of different methods and different kernels of BayesOptG](https://openreview.net/forum?id=UuNd9A6noD&noteId=FsjM9qNZ9q) for discussions on this point.
>
> > Can the Matern kernel introduced in [3] be used?
>
> We thank the reviewer for the suggestion. We can indeed introduce the Matern Kernel in our algorithm. We provided experimental results for the centrality tasks with the Matern kernel with parameter $2.5$ in Fig S1 and S2 in the rebuttal pdf.
>
> > the guarantees on the semi positive definiteness of the covariance matrix is also missing.
> > Can you please provide a bit of background on why the kernels mentioned in the paper would result in semi positive definite covariance matrices?
>
> Please see the positive semidefiniteness proof below. This will be included in the updated manuscript. In our formulation in eq 1, any map $r : \mathbb{R}  \rightarrow [0, +\infty]$ defines a valid covariance kernel.
> Indeed,
>
> $\forall \mathbf{X} \subset \mathcal{V}, k(\mathbf{X}, \mathbf{X}) = \sum_{i=1}^{\tilde{n}} r^{-1}(\lambda_i)\mathbf{u}_i[\mathbf{X}] \mathbf{u}_i[\mathbf{X}]^{T}$,
>
> where
>
> $\\mathbf{u}\_{i} [\\mathbf{X}] = [u_i[x_1], u_i[x_2], ..., u_i[x_{l}]]^{\\top}$ with $l = |X|$.
> The matrix $u_i[X] u_i[X]^{\top}$ is symmetric positive semidefinite as the outer product of one non-zero vector:
> $\forall x \in \mathbb{R}^{l}, x^{\top} u_i[X] u_i[X]^{\top}x = ||u_i[X]^{T}x||_{2}^{2} \geq 0$. As a result, our covariance matrix is symmetric positive semidefinite as the weighted sum of symmetric positive semidefinite matrices with positive coefficients.
>
> The kernels we presented in this paper correspond to a positive $r$, hence they are positive semidefinite.
>
> > The experiments section can further be improved by comparing the results with other algorithms such as spectral bandits [1], and GRUB [2].
>
> We thank the reviewer for their feedback. We agree that the settings in the spectral bandits and GRUB papers share similarities with ours. However, there are significant differences. First, both of these methods assume that the graph information is fully known and therefore does not involve graph (topology) exploration. In comparison, our approach allows for exploring a potentially unknown graph. Second, we understood both methods are global methods in the sense that the spectral bandit work requires the spectral decomposition of the graph Laplacian, while GRUB requires computing its inversion. Both operations are prohibitive for large graphs. In comparison, our method can handle large graphs (as shown in the additional experiments in Fig S3 in the PDF) due to the setup where the graph topology can be explored on-the-fly.
>
> Despite these differences, we can compare the performance of these methods to our algorithm for small and known graphs, which we will endeavour to include in the updated manuscript.
>
> > Check the captions of Figure 2 and caption of figure 3 is hard to follow.
> > Few acronyms were introduced after they were used in prior sections. ex. BA and WS in section 5.1.
> > Section A2 introduction BFS and DFS is probably jumbled.
>
> We thank the reviewer for spotting these issues, which will be rectified in an updated manuscript.

---

> > ### Comment · Reviewer_nGA9 · 2023-08-18
> >
> > I thank the authors for the response. The reviewer has no further questions.

---

> > > ### Author Response · Authors · 2023-08-21
> > >
> > > We thank the reviewer for their positive evaluation and for helping to improve our manuscript!

---

### Author Rebuttal · Authors · 2023-08-09

We thank all reviewers for their feedback. We are glad that they acknowledged the novelty (all reviewers), clarity of writing (4pXS, nGA9, GnE3), soundness (nGA9, sVur, GnE3, 4pXS), and extensiveness (GnE3, tMg5) and strength (sVur, nGA9) of experiments. We address common concerns below.

## Necessity of BO (4pXS, GnE3)

We emphasise the problems studied in Experiments indeed represent or imitate setups where BO is suitable: while they may not be expensive to evaluate *per se*, they each imitate an expensive black-box function in real-life – note that **experimenting on cheap benchmarks as a proxy to expensive, real-life tasks is done in virtually every BO paper** (e.g. evaluating synthetic benchmarks and Contimation/Pest problems in COMBO [1], the most related work, are all cheap if not instant), except that we also design these tasks and provide solutions *ourselves* because of the novel problem setup. For example:

- Identifying patient zero imitates real-life contact tracing. If executed in real life, each function evaluation requires procedures like interviews about the individuals’ travel history and people they were in contact with. This is expensive and potentially disruptive, and given limited resources, the use of a query-efficient method like BO is justified.

- Centrality maximisation & identifying influential social network nodes mirror a common task for online advertising to identify the influential users without access to the full social network information (which would be near-impossible to obtain given the number of users). Real-life social media often limits how much one may interact with their platform through pay-per-use APIs or hard limits (e.g. upper limit of views). In either case, there is a strong reason to identify the influential users in the most query-efficient manner.

## Discussions of relative strengths of different methods and different kernels of BayesOptG (nGA9, tMg5)

Many factors may affect the algorithm’s relative performance, like smoothness and noisiness of the objective functions, graph sizes, number of local minima and whether isotropy holds true (i.e., function variation in all directions is similar). We discuss each algorithm considered below (the descriptions of the baseline are provided in Appendix A.2). For ease of comparison, we also include Fig S5 in the PDF to show the methods’ rank vs the number of evaluations aggregated across all experiments.

- Random Search is simple but typically weak for larger graphs, except for very rough/noisy functions (like Ackley), or the variation in function values is generally small.
- DFS and BFS are relatively weak as they consider graph topology information only but not the node information (on which the objective function is defined) and can be sensitive to initialisation.
- Local Search is, on average, the strongest baseline, and it does particularly well on smoother functions with fewer local minima (as local search is stuck at a local minimum and requires random restarting; having few local minima reduces such occurrences). Its strength is well-documented: for example, local search on smaller-scale benchmarks in neural architecture search is competitive against state-of-the-art search algorithms [2].
- From Fig S5, BayesOptG proposed by us with *any kernel choice* outperforms baselines, but some performance differences due to the kernel choices are bound to occur. We included many possible kernel choices to inform the readers of the performance impact and encourage them to select the most suitable kernel depending on knowledge about the objective function, if any. The importance of kernel choices is well-known for *any* GP-based technique rather than being unique to our method: given that mean functions are typically set to a constant, the kernel completely determines the GP’s behaviour, its modelling effectiveness and, by extension, the effectiveness of any derived technique (e.g. BO).

As is the case for all GP-based methods, the performance is stronger when the underlying assumptions of the kernel match the actual objective function. For example, diffusion kernels work well for patient zero and team optimisation (Figs 7 & 8), as the underlying generative functions for both problems are indeed smooth (in fact, the SIR model in disease propagation is heavily connected to diffusion processes).

Diffusion without ARD further enforces isotropy, assuming the diffusion coefficient in all directions is the same, and thus typically underperforms except for team optimisation, where the generated graph is well structured (see Fig S7 in PDF) and Ackley (Fig 6 ab), which is known to be isotropic and symmetric. We recommend only if we know that the underlying function satisfies its rather stringent assumptions.

The SumInverse and DiffARD kernels are generally better, as they offer more flexibility in learning from the data; *we recommend using one of these as default if we have no prior knowledge*. The difference is that DiffARD has more learnable parameters and thus is even more flexible, but may also overfit initially when the number of observations is small – such a phenomenon can be seen in Fig S5 in the rebuttal PDF where SumInverse outperforms initially. The final kernel, Polynomial, performs weaker than the inverse; we hypothesise a possible reason is, from Table 1, a single epsilon is added to the polynomial before inversion. Even if the epsilon is chosen to be a small constant, it may still dominate the polynomial term (i.e. the learnt signal) when the latter happens to be small, and this may increase optimization difficulty (whereas for SumInverse, we add an epsilon to each polynomial term). We will thoroughly investigate this in the revised manuscript.

[1] Oh et al. (2019). Combinatorial Bayesian optimization using the graph cartesian product. NeurIPS.

[2] White et al. Local search is state of the art for NAS benchmarks. 7th ICML Workshop on Automated Machine Learning.

---

### Author Response · Authors · 2023-08-16
**Author discussion**

We thank all reviewers once again for their feedback. We believe we have largely addressed the concerns in the reviews, and we'd really appreciate it if the reviewers could look at the author feedback. We are always happy to answer any remaining questions they may have.

---

### Decision · Program_Chairs · 2023-09-21

**Decision:**

Accept (poster)

**Comment:**

This paper considers the problem of Bayesian optimization on a graph that is potentially unknown. The reviewers agree that the problem setting is interesting and could be of interest to various sections of the NeurIPS community. The proposed method appears sound, and the experimental results seem to support its benefits compared to other methods in the literature.